# Hydrograph separation: an impartial parametrization for an imperfect method

Antoine PELLETIER[1,2] and Vazken ANDRÉASSIAN[2]
[1]École des Ponts ParisTech, Champs-sur-Marne, France
[2]Irstea, UR Hycar, Antony, France
**Correspondence:** Antoine Pelletier (antoine.pelletier@irstea.fr)

**Abstract.** This paper presents a new method for hydrograph separation. It is well-known that all hydrological methods aiming at separating streamflow into baseflow and quickflow present large imperfections, and we do not claim to provide here a perfect solution. However, the method described here is at least (i) impartial in the determination of its two parameters (a quadratic reservoir capacity and a response time), (ii) coherent in time (as assessed by a split-sample test) and (iii) geologically coherent (an exhaustive validation on 1,664 French catchments shows a good match with what we know of France's hydrogeology). Last, an R package is provided to ensure reproducibility of the results presented.

## 1 Introduction

Hydrograph separation and the identification of the baseflow contribution to streamflow is definitely not a new subject in hydrology. This age-old topic (Boussinesq (1904); Horton (1933); Maillet (1905)) is almost as universally decried as it is universally used. Indeed, two adjectives appear repeatedly in hydrology textbooks: *artificial* and *arbitrary* (see. e.g., Linsley et al. (1975); Réméniéras (1965); Roche (1963); Chow (1964)). Hewlett and Hibbert (1967) – the famous forest hydrology precursors – even added *desperate*, where Klemes (1986) compared the hydrograph separation procedures with the astronomical epicycles (i.e., the absurd trajectories that had been invented to maintain the geocentric theory before the time of Copernicus and Galilei).

To assess baseflow, direct measurement is generally impossible, because aquifer–river exchanges are usually continuous processes that take place all along the stream network. Proxy approaches involving chemical tracer-based procedures are efficient but need chemical data and involve their load of assumptions. Most approaches rely on solving an *inverse problem*, i.e, reckoning the quantitative causes (here baseflow and quickflow) of an observed physical phenomenon – total runoff. This procedure is very common in hydrology and it is reasonably feasible when the variable can be measured and a calibration procedure can be implemented; but here again, the non-measurable character of baseflow renders the question difficult.

It is perhaps impossible to propose a physically based baseflow separation procedure (just because of the multiplicity of flow paths that make the procedure fundamentally equifinal), and we will not argue on this point. But we believe that even the imperfect conceptual–mathematical–empirical methods in use could receive an unarbitrary, impartial, repeatable parameterization that could be used as a general-purpose study tool over large catchment sets.

This paper focuses on a hydrograph separation method that is based only on quantitative streamflow data and climate descriptors and does not require *a priori* physical parametrization, presented in section 3. The originality of this method lies in its parametrization strategy, which we discuss in detail in section 3.2. The application of this procedure to a set of 1,664 catchments, its geological coherence and its stability in time are presented in section 4.

## 2 Hydrograph separation: a short review

Existing procedures for hydrograph separation and baseflow assessment can be divided into three categories (Gonzales et al. (2009)), according to their level of *physicalness*: (i) physical/chemical methods are based on the differences – of composition or temperature, for instance – between groundwater and surface water; (ii) numerical/empirical methods do not attempt any kind of hydrological representation or modelling and use signal processing tools to alter streamflow hydrographs; (iii) conceptual methods conceive the aquifer as a reservoir that empties itself during recessions. The following section paints a quick picture of these three categories, before concluding about their shortages and drawbacks that led to the development of a new method.

### 2.1 Chemicophysical methods

Chemicophysical methods are based on the fact that total flow is a *mix* of baseflow water and surface flow water, with different characteristics (Pinder and Jones (1969)). By knowing the latter, it is possible to revert mixing equations to get the relative contribution of each flow source, using a simple mass-balance approach. Chemicophysical methods are powerful and they provide a lot of information where surface water and groundwater quality data are available and where the hydrochemical configuration allows for rather strong hypotheses: consistency in space and time of water characteristics and the assumption that the hydrological system is conservative – to avoid any unknown water inflow.

The most common methods of this family are tracer-based baseflow separations. A passive tracer whose concentration is different in groundwater and surface water – for instance, isotopes such as deuterium or $^{18}$O, ions (carbonates, calcium, magnesium, sulphates) from dissolved minerals or silica – is used as a proxy for mass balance assessment, at each time step of available data, which allows for a dynamic real-time baseflow decomposition. Other methods rely on conductivity, pH, or temperature, which can be easier to measure in situ.

There are two main drawbacks of chemicophysical methods. First, the uncertainties that are inherent to them – there are measurement errors, concentrations can have spatial and temporal heterogeneity, even within the same rainstorm, and the passiveness of tracers may not be a valid hypothesis even in short events; Kirchner (2019) proposed a statistical tracer-based separation to replace this mass balance hypothesis, using a regression between baseflow and tracer concentrations in streamflow and rainfall. Second, the implementation of these methods on catchments is limited by the large number of data needed. Continuous groundwater and surface water quality data are seldom available except for specially instrumented catchments (Gonzales et al. (2009)). Even though tracer-based methods can provide useful and valuable information on dominant processes occurring inside a catchment, their application for routine analysis is not conceivable on a large number of gauging stations.

### 2.2 Empirical and numerical methods

Several classic baseflow-separation methods are not based on hydrological considerations, but rely on processing the hydrograph as a signal. Most of these methods are based on the hypothesis that the transfer time of surface runoff is much shorter than that of groundwater and that this time is relatively constant between rainfall events. The first methods of this type were graphical. After identifying peaks along the hydrograph and estimating the surface runoff time constant – let us say $N$ days – peaks are cut by drawing a straight line between the beginning of each peak and the point $N$ days after (Linsley et al. (1975); Gonzales et al. (2009)). This simple approach has been honed to take into account consecutive precipitation events or aquifer recharge during rainfall, but it remains quite subjective, as the graphical processing is supposed to be done by hand and is difficult to automatize. Streamflow under the constructed dividing line is assumed to be baseflow, while the remaining peaks are surface flow.

Within the same set of hypotheses, low-pass numerical filtering of the hydrograph has been used as a baseflow separation method. As in electronic signal filtering, the highest frequencies of the signal are dropped and the low ones are kept. Fourier transform-based filters implement this framework, but they are intended for data with stationary periodic processes, whereas hydrological signals operate on a very large range of time scales. Even *quick* components of streamflow have low-frequency Fourier terms (Duvert et al. (2015); Labat (2005)). Classical electronic low-pass filters are therefore not adapted to hydrograph separation. In line with graphical methods, a local minimum filter on a sliding window – whose length is basically the time scale of baseflow – has been developed, as well as recursive digital filters with low-pass properties. Transformations can be applied on the streamflow time series before numerical filtering, in order to improve the filter's efficiency (Romanowicz (2010)). Finally, artificial neural networks have been employed as data-driven models for hydrograph separation (Taormina et al. (2015); Jain et al. (2004)).

Numerical methods all need one or several parameters, with or without hydrological meaning, which have to be valued before processing: time interval width, coefficients of a linear filter, formula of the transformation function, etc. Even though authors generally give advice about the possible values of these parameters, determining them involves adding new hypotheses for each catchment examined (Eckhardt (2008, 2005)). In particular, quantitative results of the filtering change with the value of the parameters. Although the shape of the hydrograph separation can be eye-pleasing, it is basically impossible to draw any conclusion from it.

### 2.3 Conceptual methods

In addition to chemicophysical and numerical methods, another way of separating hydrographs has been developed, starting from the idea that aquifers can be represented as conceptual reservoirs, whose outflow is baseflow. Such an approach comes from the analysis of long recession curves and the inference of depletion laws, which rule streamflow during long, dry periods of low flows. By knowing the outflow law of the conceptual reservoir, assuming that its input is zero, it is possible to get the full recession curve until the next significant rainfall event (Coutagne (1948); Wittenberg (1994)). In practice, various types of

reservoirs are implemented in this type of model: linear, quadratic and exponential are the most common (Bentura and Michel (1997); Michel et al. (2003)).

It is possible to calibrate such a model by fitting it on *long recession curves*, but the latter have to be defined, for instance, through a threshold on streamflow – considering that only low flows correspond to recessions – or on its derivative, possibly with a smoothing filter – such as moving average or sliding window minimum – in order to eliminate noise in the signal. Such a sampling of the hydrograph can be difficult to make without arbitrariness, either in the value of thresholds or in smoothing filters. Existing studies generally focus on a particular recession, for instance, along a drought in which streamflow is known to have been measured well (Wittenberg and Sivapalan (1999); Wittenberg (1994); Coutagne (1948)). It may be impossible to find such a drought to respect the zero-input hypothesis – in Western Europe, for instance, rainstorms at the end of summer often disrupt the low-flow signal.

To filter the whole hydrograph with this conceptual idea of baseflow, a backward filter was developed by Wittenberg (1999). The hydrograph is traced back with reservoir recession curves; the level of the reservoir increases as depletion is followed backward and reinitialized when the peak is reached. Some arbitrariness remains in the value of the reservoir parameter; moreover, whereas this method yields good-looking hydrograph separations with wide time steps, baseflow may not be smooth enough under flood peaks: a further smoothing is thus needed and constitutes, here again, a source of partiality.

### 2.4 Elements of comparison and coupling between methods

Although it is impossible to perform an absolute evaluation of a hydrograph separation method – since baseflow cannot be measured and compared with a simulated value – several comparative studies have challenged results from various methods. Chemicophysical and non-calibrated graphical/empirical algorithms – i.e., without parameters or whose parameters are determined *a priori* and do not vary between catchments – have given very different results (Kronholm and Capel (2015); Miller et al. (2015); Lott and Stewart (2016)). It is therefore difficult to use non-calibrated empirical or graphical hydrograph separation methods as general-purpose analysis tools.

To remedy this, several studies use parametric graphical/empirical methods and calibrate them with chemicophysical data (Saraiva Okello et al. (2018); Miller et al. (2015); Chapman (1999)). It allows one to extend the hydrograph separation on time periods where chemicophysical data are not available, or to perform temporal downscaling – when daily streamflow data or monthly tracer concentrations are available, for instance. Such calibrated parametric algorithms have given satisfying results. Chapman (1999) highlighted that adding parameters to a hydrograph separation algorithm does not necessarily improve performance, since the algorithm becomes harder to calibrate.

When performing a large-scale baseflow analysis over a territory, it is possible to extend calibrated parameters at gauged catchments to ungauged ones, through regionalization models. Singh et al. (2019) performed such a study over the entire territory of New Zealand. We decided to limit ourselves to gauged catchments for this work.

Hydrology and Earth System Sciences
EGU
Discussions

### 2.5 Synthesis and guidance for this work

Hydrograph separation into baseflow and quickflow relies on a set of hypotheses about the general behaviour of a river basin. Streamflow in a river is seen as having two origins, groundwater – the sum of contributions of various aquifers – and surface water, made of surface runoff and subsurface interflow, i.e., water that does not stay too long underground. Surface water response to climatic events is much quicker than that of groundwater and this speed difference is time-coherent all along the hydrograph. All this eliminates strongly karstic catchments, as the response of karst systems can be almost as quick as surface runoff.

Most of the methods above rely on one or several parameters whose value has a strong influence on hydrograph separation; as it is impossible to calibrate this (or these) parameter(s) on the basis of a measurement, some physical hypotheses must be made on the desirable properties of baseflow, in relation to the memory of the catchment. Non-calibrated algorithms are not usable as general-purpose analysis tools and existing calibration procedures rely on the availability of chemicophysical data. In this work, we propose a calibration procedure of a conceptual hydrograph separation method that relies only on hydroclimatic data: streamflow, rainfall and temperature.

Frugality and objectivity do not only result in a small number of parameters. The choice of modelling options is also a source of arbitrariness and useless complication. A modelling alternative can, moreover, be solved only through a personal choice of the modeller; one way to build a trustworthy framework is to make it of well-known and well-validated elements that have proven themselves relevant in lumped conceptual hydrology. Beyond the complexity of elements, there is the complexity of the modelling chain itself. The simpler the model is, the more readable and thus the less questionable it is; in this work, it was considered that a baseflow separation process should be much simpler than, for instance, a lumped conceptual model designed for flow simulation.

## 3 Proposed methodological framework

### 3.1 Digital filtering based on conceptual storage

In this paper, we postulate that the memory and smoothing effect of a catchment – which underpins the concept of baseflow – can be represented by a conceptual reservoir, whose outflow will represent the groundwater contribution to streamflow, which we will assimilate into baseflow. To be applied in practice, this postulate must be complemented by an answer to the two following questions: what should the input to the reservoir be? What should the relationship between the level of the reservoir and its outflow be?

#### 3.1.1 Discharge as a proxy of recharge

The first question is: "What should refill the conceptual reservoir?"; i.e., "What recharges the aquifer with water that will be baseflow afterwards?". The part of rainfall that does not contribute to surface runoff or to evaporation is generally named *recharge*, as it is thought to feed groundwater storage. It is common to estimate the recharge function through a surface water

budget model, which computes it from climatic information (rainfall, evapotranspiration, temperature, etc.) as it is done more or less complexly in common rainfall–runoff models or in soil–vegetation–atmosphere schemes. But using such production modules would introduce additional hypotheses and parameters inside the baseflow separation method. Instead, we looked for a proxy of recharge, which would represent at least a general yearly behaviour on a rainfall-dominated basin. During high-flow periods, recharge is high, whereas during low-flow periods, most or all the available water goes to evaporation, and thus recharge is low.

In this paper, we make the hypothesis that the best candidate for a proxy of recharge is a linear fraction of total flow itself. It is quite well-correlated with the behaviour of recharge given above. In addition, it is available without a further model or hypothesis. But using a linear fraction of total flow raises an issue: if $x$ % of total flow is poured into the reservoir and baseflow – which represents on average $y$ % of total flow – is taken out of the reservoir, then the water budget of the filtering process will only be balanced if $x = y$. The parametrization strategy takes this problem into account; it is presented in section 3.2.

#### 3.1.2 Reservoir filtering

The filtering part of the method is a conceptual reservoir, whose inflow is a fraction of the total observed flow at the gauging station and the outflow is regarded as baseflow. The content of this reservoir is managed by an outflow function and a continuous differential equation: since data are obviously available as discrete time series with a time step $\Delta t$, it is necessary to discretize this equation to set the numerical filtering algorithm.

The following notations will be used henceforth. All variables are expressed as intensive values – i.e., as depths in millimetres for precipitation, flows, reservoir storage – in order to avoid conversions with the basin area.

- $Q(t)$: total measured flow in mm per $\Delta t$ ;
- $R(t)$: baseflow in mm per $\Delta t$ ;
- $P(t)$: rainfall in mm per $\Delta t$ ;
- $V(t)$: reservoir storage in mm ;
- $\beta$: fraction of total flow that enters the conceptual reservoir.

According to the definition of a reservoir, $V(t)$ is governed by the following system of equations:

$$\frac{\mathrm{d}V}{\mathrm{d}t} = \beta Q(t) - R(t) \quad (1)$$

$$R(t) = f\left(V(t), S\right) \quad (2)$$

$f$ is the outflow function of the reservoir, with a parameter $S$ – whose dimension depends on the formula of $f$. Many functions can be imagined, but the most common ones are the following (Brodie and Hosteletler (2005)):

- linear reservoir: $f(V(t),S) = \frac{V(t)}{S\Delta t}$ ;
- quadratic reservoir: $f(V(t),S) = \frac{V(t)^2}{S\Delta t}$ ;
- exponential reservoir: $f(V(t),S) = R_0 \exp\left(\frac{V(t)}{S}\right)$.

Linear reservoir has been widely used because of computation ease and direct analogy with linear filtering systems: the reservoir can be, for instance, assimilated into a low-pass electronic filter – the comparison with a capacitor and a resistor is quite straightforward. On the contrary, computations with the two other functions involve solving a non-linear differential equation, which removes the analogy with classical linear filters. Notwithstanding these practical considerations, groundwater discharge is fundamentally a non-linear phenomenon. Several authors (Coutagne (1948); Wittenberg (1994, 1999); Bentura and Michel (1997)) have argued that from a general-purpose point of view, a quadratic reservoir is the most adapted function. It has been derived from statistical studies of various catchments, together with theoretical studies of springs and unconfined aquifers. Therefore, a quadratic reservoir is chosen for this work.

In order to discretize the filter, we define discrete versions of continuous variables cited above. If $X(t)$ is continuous, for a discrete time step $t$, we define $X_t = X(t)$. To integrate equation 1, the reverse process is performed: $Q(\tau)$ is simply built from measured $Q_t$ through a sum of Dirac distributions. It simply means that at each time step $t$, $\beta Q_t$ is poured into the reservoir at the beginning of the time interval $[t\ ;\ t+\Delta t]$:

$$Q(\tau) \simeq \sum_t Q_t \delta_t(\tau)$$

Although it is used in most conceptual hydrological models, this way of integrating differential equations for a discrete hydrological model could be criticized (Fenicia et al. (2011); Ficchì (2017)) because the explicit Euler integration scheme is not stable for solving many ordinary differential equations. For instance, it can lead to diverging numerical solutions while analytical ones are supposed to be bounded. We find it acceptable to use this explicit integration scheme for the sake of simplicity, given that divergence of the result is controlled by the reservoir update detailed in section 3.1.3.

Thus, $V(t^-) = V_t$ and $V(t^+) = V_t + \beta Q_t \Delta t$. With a quadratic reservoir function, equation 1 can now be written on the time interval $[t^+\ ;\ (t+\Delta t)^-]$:

$$-\frac{\mathrm{d}V}{V^2} = \frac{\mathrm{d}t}{S\Delta t} \tag{3}$$

After integration, the discrete recursive equation is:

$$V_{t+\Delta t} = \frac{V_t + \beta Q_t \Delta t}{1 + \frac{V_t + \beta Q_t \Delta t}{S}} \tag{4}$$

$R_t$ is the total outflow during the time interval $[t\ ;\ t+\Delta t]$. From another integration of equation 1, a balance equation is obtained: $V_{t+\Delta t} = V_t + \beta Q_t \Delta t - R_t \Delta t$. Therefore, using equation 4, the following formula for $R_t$ is obtained:

$$R_t = \frac{1}{\Delta t} \frac{(V_t + \beta Q_t \Delta t)^2}{S + V_t + \beta Q_t \Delta t} \tag{5}$$

Note that these equations have a physically coherent behaviour: $V_t$ and $R_t$ are always positive; without inflow, equation 4 becomes a depletion equation, which converges to zero; the reservoir empties itself all the faster as its level is high ; $S$ acts as a capacity, i.e., at the end of a time step, the level of the reservoir is always lower than $S$.

#### 3.1.3 Reservoir update

The filtering process detailed above raises an issue: baseflow is by definition a fraction of total flow; yet, equation 5 does not guarantee that $R_t$ is lower than $Q_t$. Conceptual rainfall–runoff models classically rely on evaporation or on a leakage function to empty their reservoirs. Here, for the sake of simplicity, it was decided to perform a simple update of the reservoir level, when the outflow is greater than the measured flow. Although it may seem quite brutal, it must be seen as a readjustment of the model after a flood peak, in order to fit the observed flood recession. Since part of streamflow is only used as a proxy of recharge – and is therefore not the recharge itself – it is an imperfect and conceptually questionable input data for the model and thus a correction process within the latter is not a surprise.

If $R_t$ is computed as greater than $Q_t$, the level of the reservoir $V_t$ needs to be updated so that $R_t = Q_t$, which leads through a simple balance of the reservoir to $V_t = V_{t+\Delta t}$. Equation 6 then gives the update function, by solving a quadratic polynomial equation:

$$V_t = \frac{1}{2} Q_t \Delta t \left( \sqrt{1 + \frac{4S}{Q_t \Delta t}} - 2\beta + 1 \right) \quad (6)$$

Here again, $V_t$ is always non-negative and strictly lower than $S$. The formula above artificially creates a numerical exception when $Q_t = 0$, which must be taken into account in the implementation of the algorithm, with $V_t = 0$.

This update process takes water out of the reservoir when computed baseflow is too high; but do we not need to correct baseflow when it is too low? Indeed, quickflow is supposed to be zero during long rainless, dry periods. For the sake of simplicity, a straightforward hypothesis is added: baseflow must be equal to total flow at least once in a hydrological year, when measured streamflow reaches its yearly minimum. The value of the latter can be affected by measurement errors, but it is hard to take it into account in a general-purpose analysis tool: the structure of the error is determined by the idiosyncrasies of each catchment. Moreover, the two ways of updating the reservoir helps the model to forget about previous errors; therefore, for the sake of simplicity, we prefer not to add another hypothesis about the low-flow measurement uncertainty.

A test study is performed on a set of catchments in continental France: thus, the low-flow hydrological year is taken from the 1st of April to the 31st of March, as streamflow minima do not generally occur during spring. The $Q_t$ time series is then sliced into hydrological years, and for each of them, a minimum value and its index are computed. If the series is $n$ hydrological years long, it creates a set of $n$ points where reservoir level is updated through equation 6, so that $R_t = Q_t$.

This second update process adds water to the reservoir; it is thus possible to balance the updates of the reservoir and, thereby, to balance the total water budget of the filtering process. If $\bar{Q}$ and $\bar{R}$ are the means of time variables $Q_t$ and $R_t$, water budget is given by equation 7.

$$\beta\bar{Q} = \bar{R} = BFI \times \bar{Q} \tag{7}$$

*BFI* is *baseflow index*, namely, the long-term ratio of the computed baseflow to total measured streamflow – i.e., $BFI = \frac{\bar{R}}{\bar{Q}}$. Equation 7 gives a simple parametrization constraint to ensure a balanced water budget: $\beta = BFI$.

#### 3.1.4 Initialization

Our algorithm does not need a long warm-up period, as it exists for rainfall–runoff models. Since the algorithm presented above contains only one state and involves a regular update with observed data, this update erases the memory of the past states of the model. Therefore, a simple initialization approach is possible. Several methods were tested and the best compromise between simplicity and robustness was adopted. The reservoir level is initialized through equation 6, using mean flow over a small time period – five time steps – at the beginning of the flow time series. To avoid any overestimation of baseflow at the beginning of the hydrograph, computations were started far from any flood events: the 1st of August is a convenient date for continental mainland France since no French river has major high-flow episodes in mid-summer.

#### 3.1.5 Synthesis of the algorithm

Finally, the filtering algorithm can be summarised with the pseudo-code detailed in algorithm 1. $f_{up}(Q_t)$ is the update function of equation 6, $f_{rec}(V_t, Q_t)$ is the recursion function of equation 4, $n$ is the number of time steps and $YM$ is the set of indexes of yearly minima. Figure 1 synthesises the principle of the algorithm.

### 3.2 Parametrization strategy

As stated above, the difficulty of parametrization comes from the fact that baseflow cannot be measured. In the absence of measurement, we need to make an hypothesis concerning a desirable property of the computed baseflow, in order to be able to look for the parameter set that will best respect this property. Since baseflow accounts for the slow component of total flow, coming out from the storage parts of a river basin, we make the following hypothesis: baseflow should be correlated with past climatic events that have filled or emptied this storage ; and since it is supposed to be slow, it should be correlated with events that happened quite a long time ago. The calibration criterion has to be a simple quantification of this idea, which corresponds to the concept of *recharge* stated earlier: a proxy of past recharge must be found.

The filtering algorithm detailed above depends on two parameters: capacity $S$ of the reservoir and scaling factor $\beta$. Yet, with the water balance condition given by equation 7, it is possible to remove one degree of freedom: for a given value of $S$, function $\beta \mapsto BFI(S,\beta) - \beta$ is injective; therefore, there is a unique value of $\beta$ that satisfies $BFI(S,\beta) = \beta$ and the $\beta(S)$ function can be identified.

Hydrology and Earth System Sciences
EGU
Discussions

**Algorithm 1** Reservoir filtering algorithm

**Initialization**

$V_1 \leftarrow f_{up}(\textsc{Mean}(Q_i, i \in [1;5]))$

**Recursion loop**

**for** $i = 0$ to $n$ **do**

  $V_{i+1} \leftarrow f_{rec}(V_i, Q_i)$

  $R_i \leftarrow Q_i + \Delta t(V_i - V_{i+1})$

  **if** $R_i > Q_i$ or $i \in YM$ **then**

    $R_i \leftarrow Q_i$

    $V_i \leftarrow f_{up}(Q_i)$

    $V_{i+1} \leftarrow V_i$

  **end if**

**end for**

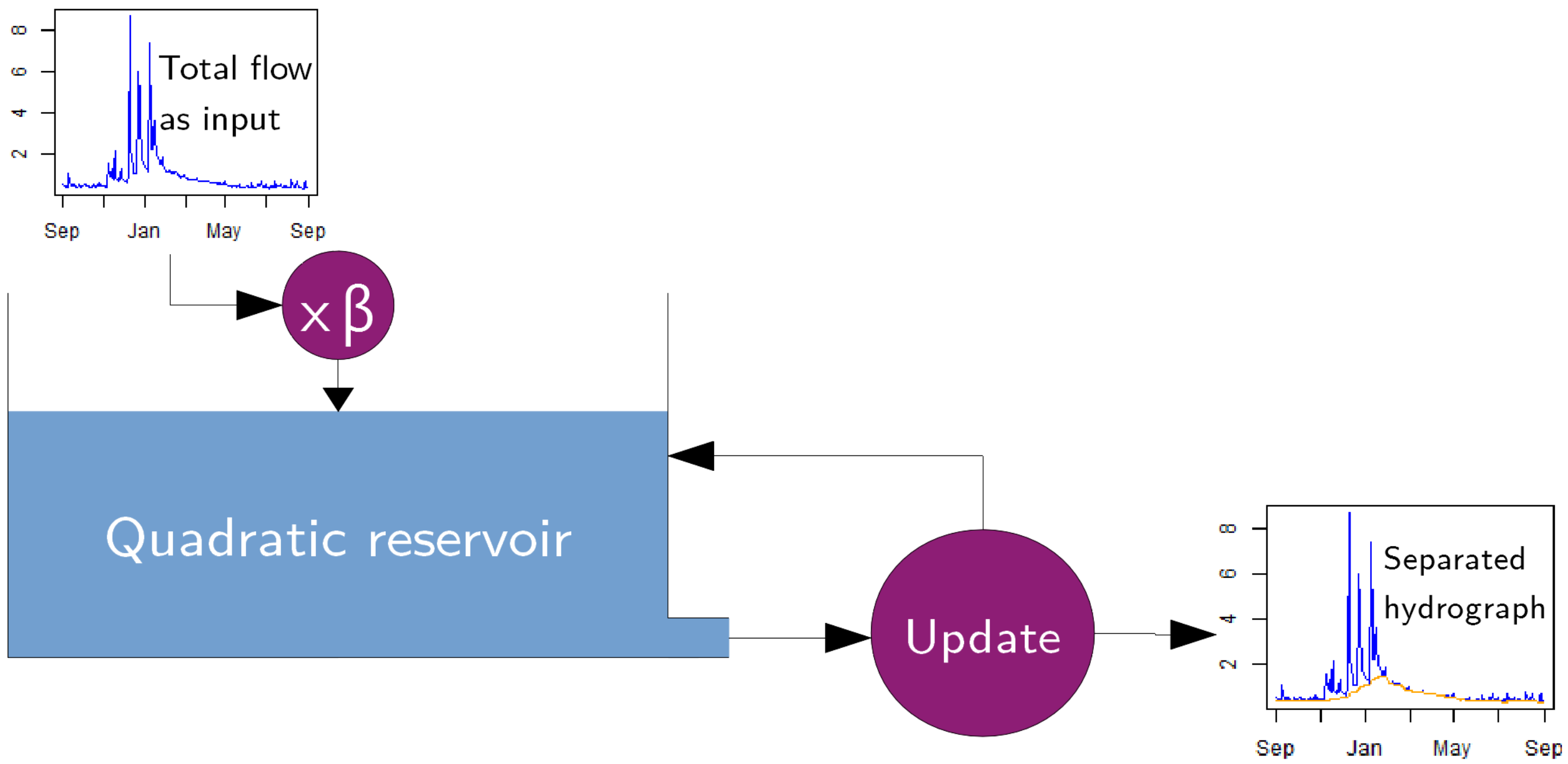

**Figure 1.** Synthesis of the filtering algorithm

After eliminating one degree of freedom, the correlation hypothesis stated above can be converted into a criterion: it computes Pearson's correlation between the computed baseflow time series – which depends on the reservoir's parameter, $S$ – and cumulated effective rainfall on the $\tau$ days before each time step. The first $\tau - 1$ values of the baseflow time series thus do not influence the criterion, since the first $\tau - 1$ values of the cumulated effective rainfall time series are missing values and,

therefore, they are eliminated when computing Pearson's correlation; this limits the effect of model initialization. If $P_{eff,t}$ is the daily effective rainfall value on day $t$, the criterion is written:

$$C(S,\tau) = \text{Corr}\left(R_t(S), \sum_{u=t-\tau+1}^{t} P_{eff,u}\right) \quad (8)$$

This correlation must be maximized to yield two pieces of information: the optimal value $S_{opt}$ of the capacity of the conceptual reservoir, which gives a hydrograph separation; a time response $\tau_{opt}$ of the river basin, which can be compared with other pieces of knowledge about its behaviour, in order to assess the separation method's relevance.

As far as the optimization process is concerned, it is necessary to define search intervals for $S$ and $\tau$. For $S$, values lower than 1 mm were not observed during tests and led to computation issues, thus the lower bound can be set at 1 mm; on the other side of the search interval, values around $10^6$ mm – 1 km – were found during tests, without leading to incongruities in baseflow separation; therefore, it is suggested to set the upper bound to $2.10^6$ mm. For $\tau$, typical values are between 3 months and 1 year: the search range is then set to $[5;1825]$ days. Note that during tests, $\tau$ optimal values were easier to find than $S$ values.

Effective rainfall was computed with the Turc–Mezentsev formula (Turc (1953); Mezentsev (1955)), with a quadratic harmonic mean between potential evaporation (PET) and rainfall, i.e.:

$$P_{eff} = P_{tot}\left(1 - \frac{1}{\sqrt{1+\left(\frac{P_{tot}}{PET}\right)^2}}\right)$$

$P_{tot}$ is daily total rainfall and $PET$ is daily potential evapotranspiration, computed with Oudin's formula (Oudin (2004); Oudin et al. (2005)): effective rainfall is thus computed at a daily time step and aggregated to get the criterion defined by equation 8. By way of comparison, alternative ways of computation were tried: Penman–Monteith formula for PET (Monteith (1965)) and simple screening of rain by PET for effective rainfall – i.e., $P_{eff} = \text{Max}(P_{tot} - PET, 0)$. No major difference was found in the results.

Therefore, we found an effective parametrization strategy for choosing the three used parameters $\beta$, S and $\tau$. $\beta$ is first linked to S through the budget condition given by equation 7, which removes one degree of freedom; then, the two remaining parameters S and $\tau$ are chosen by optimizing the correlation criterion given by equation 8.

### 3.3 Catchment dataset

Several hydrographic regions of mainland France are influenced by large aquifers, which bear memory of past climatic events and have a significant contribution to river flow. In the Paris basin, the Chalk aquifer – composed of Late Cretaceous formations – has a significant connection with many rivers in the Seine and the Somme basins, which notably led to major groundwater-induced floods after the extremely wet years of 1999 and 2000 (Pinault et al. (2005); Habets et al. (2010)). The Loire basin is linked to the Beauce aquifer – mostly made of tertiary layers – whose situation under a large agricultural plain induces water

management issues that affect Loire low-flows. The international Rhineland aquifer, part of which extends to the Alsace, also plays a major role in the hydrology of the Rhine.

Tests were performed on a set of 1,664 catchments that were selected on the basis of diversity of area, climate, hydrological regime and completeness of the time series in the period 1967–2018. Streamflow data are from the national database Banque Hydro (SCHAPI (2015); Leleu et al. (2014)), where measurements regarded as unreliable were converted into missing values. Climatic data are from Safran reanalysis (Vidal et al. (2010)), developed by Météo France and aggregated for each catchment, to get a lumped climatic series of rainfall and PET. The combination of the Safran and Banque Hydro data is stored in the HydroSafran database, maintained and hosted by Irstea (Delaigue et al. (2019)).

Gaps in the streamflow time series were filled using simulated flows from the daily lumped rainfall–runoff model GR4J (Perrin (2002); Perrin et al. (2003)), calibrated on the whole period 1968–2018 – we make the hypothesis that hydrological characteristics of catchments are stationary all along this period. The chosen catchments have more than 20 complete years – a hydrological year is regarded as complete when there is no gap longer than a week and the total proportion of missing values is lower than 5 % – and the filling simulation model GR4J has a good performance, with a Nash–Sutcliffe criterion higher than 0.7. All computations were performed with R and airGR package for specific hydrological processing (Coron et al. (2017, 2019)).

**Table 1.** Geographic characteristics of catchment dataset

| | **Outlet altitude (m)** | **Average altitude (m)** | **Average hydraulic slope** | **Area (km²)** |
|---|---|---|---|---|
| *Minimum* | 0.50 | 22.3 | 0.01 | 1.86 |
| *First quartile* | 72.0 | 181 | 0.05 | 95.3 |
| *Median* | 169 | 341 | 0.08 | 229 |
| *Mean* | 237 | 472 | 0.12 | 1,828 |
| *Third quartile* | 304 | 664 | 0.15 | 716 |
| *Maximum* | 2,154 | 2,871 | 0.67 | 117,183 |

**Table 2.** Hydrological characteristics of catchment dataset. Aridity index is defined as the quotient of average rainfall by average PET. NB: Precipitation yield greater than 100% was encountered in a small karstic catchment where supplementary water is brought by a major resurgence.

| | **Annual streamflow (mm)** | **Annual precipitation (mm)** | **Precipitation yield (%)** | **Aridity index** |
|---|---|---|---|---|
| *Minimum* | 59 | 617 | 5 | 0.75 |
| *First quartile* | 248 | 849 | 29 | 1.20 |
| *Median* | 355 | 960 | 37 | 1.43 |
| *Mean* | 446 | 1,035 | 40 | 1.59 |
| *Third quartile* | 543 | 1,137 | 48 | 1.75 |
| *Maximum* | 2,300 | 2,219 | 164 | 6.05 |

Hydrology and Earth System Sciences
EGU
Discussions

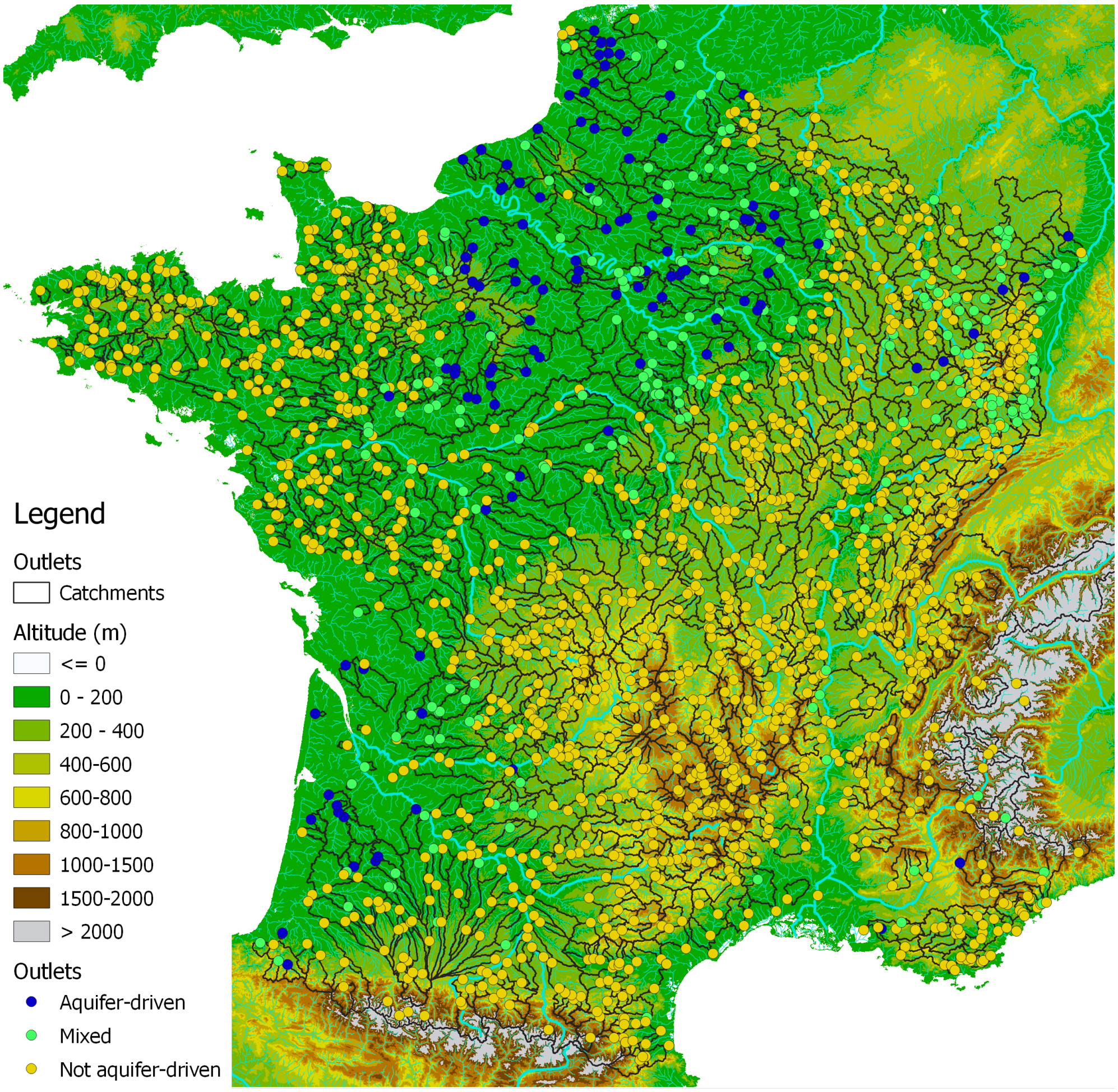

**Figure 2.** Map of the catchments in the dataset. Dots represent the outlet of each basin, with different colours according to the type of aquifer influence used in figure 8.

Tables 1 and 2 show some characteristics of the catchment dataset: it is representative of the climatic and landscape diversity in mainland France. We chose basins with a large range in size, from very small (less than 2 km$^2$) to very large (117,000 km$^2$). A quick geological analysis of the 1,664 catchments was also performed based on the national hydrogeological map by Margat (1980); it highlighted that 1,310 catchments are not very likely to be influenced by aquifers, since they are on impermeable formations, such as igneous or metamorphic rocks or clay; 122 catchments have more than 90 % of their surface on sedimentary formations that are likely to carry large aquifers; and 232 catchments have a mixed configuration. Figure 2 shows the geographic repartition of the catchment set.

For each catchment of the dataset, a two-variables grid search was performed to find the optimum of the criterion. Hydrograph separation was then computed with the parameters obtained.

## 4 Results

### 4.1 Criterion surface plots

We already mentioned that, for each catchment, there is a perfect bijection between values of the reservoir parameter $S$ and values of the BFI: the BFI is a decreasing function of $S$. The BFI values are limited: in fact, when $S$ tends toward infinity, the filter behaves like a minimum finder and baseflow is constant, equal to the minimum flow; thus the BFI is always greater than $Q_{min}/Q_{avg}$, the ratio of minimum flow over mean flow. The BFI also has an upper bound, due to the numerical impossibility of getting a baseflow equal to total flow with the filter; therefore, a range of possible BFIs was computed for each catchment and the optimization criterion can be seen as a function of the BFI and $\tau$, defined for BFI values within this range. This function was then plotted on a contour plot to ensure reasonable optimization configuration.

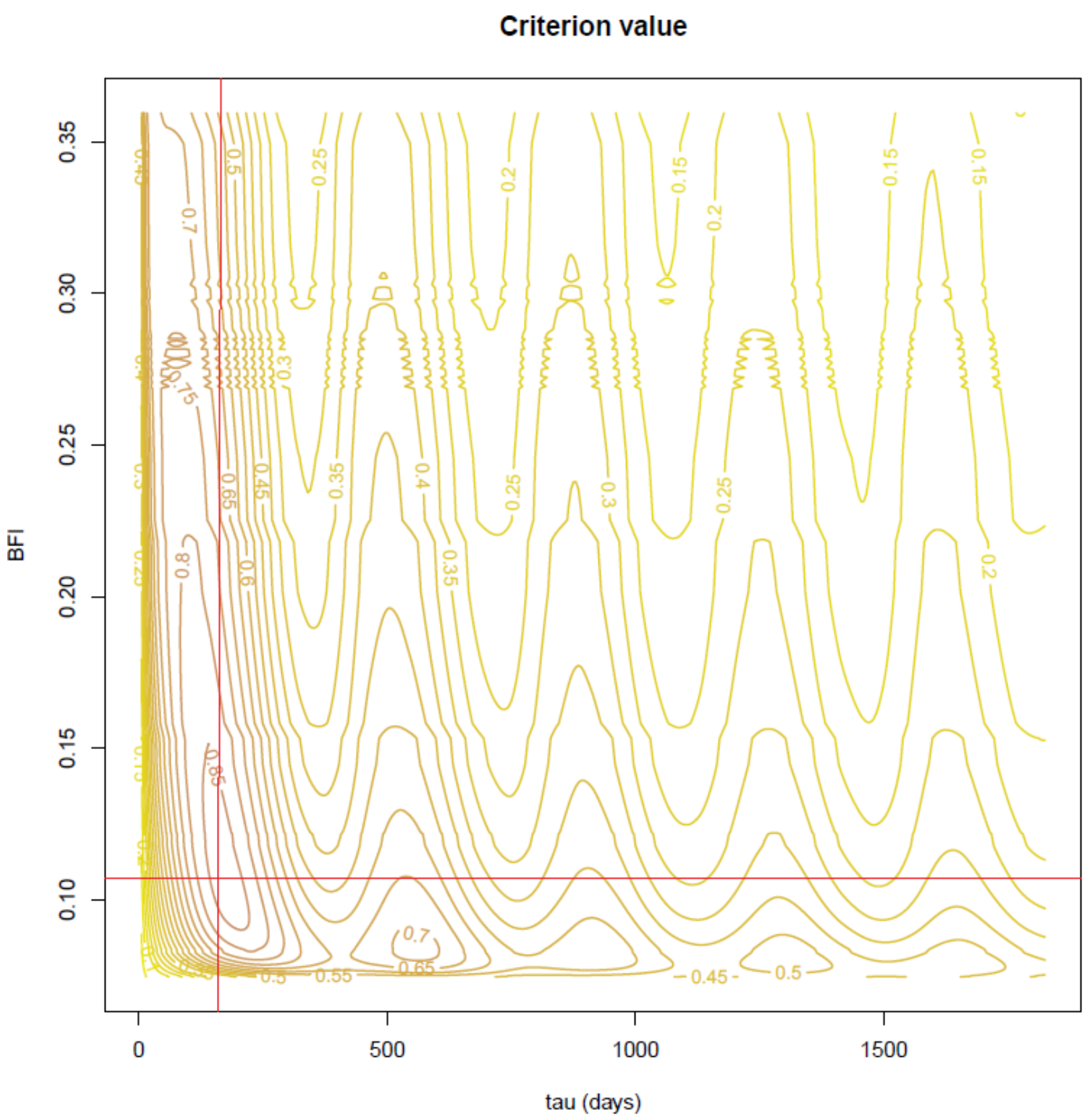

**Figure 3.** Surface plot of optimization criterion for the Vair river in Soumousse-sous-Saint-Élophe. Red cross indicates the numerical optimum.

Two examples of surface plots are presented in figures 3 and 4. The Vair river in Soumousse-sous-Saint-Élophe is a medium-sized basin dominated by impervious sedimentary formations. It is thus likely to have low BFI and response time. The criterion optimization is clearly unequivocal: the optimal point – which gives a correlation of $0.86$ – has a difference greater than $0.15$

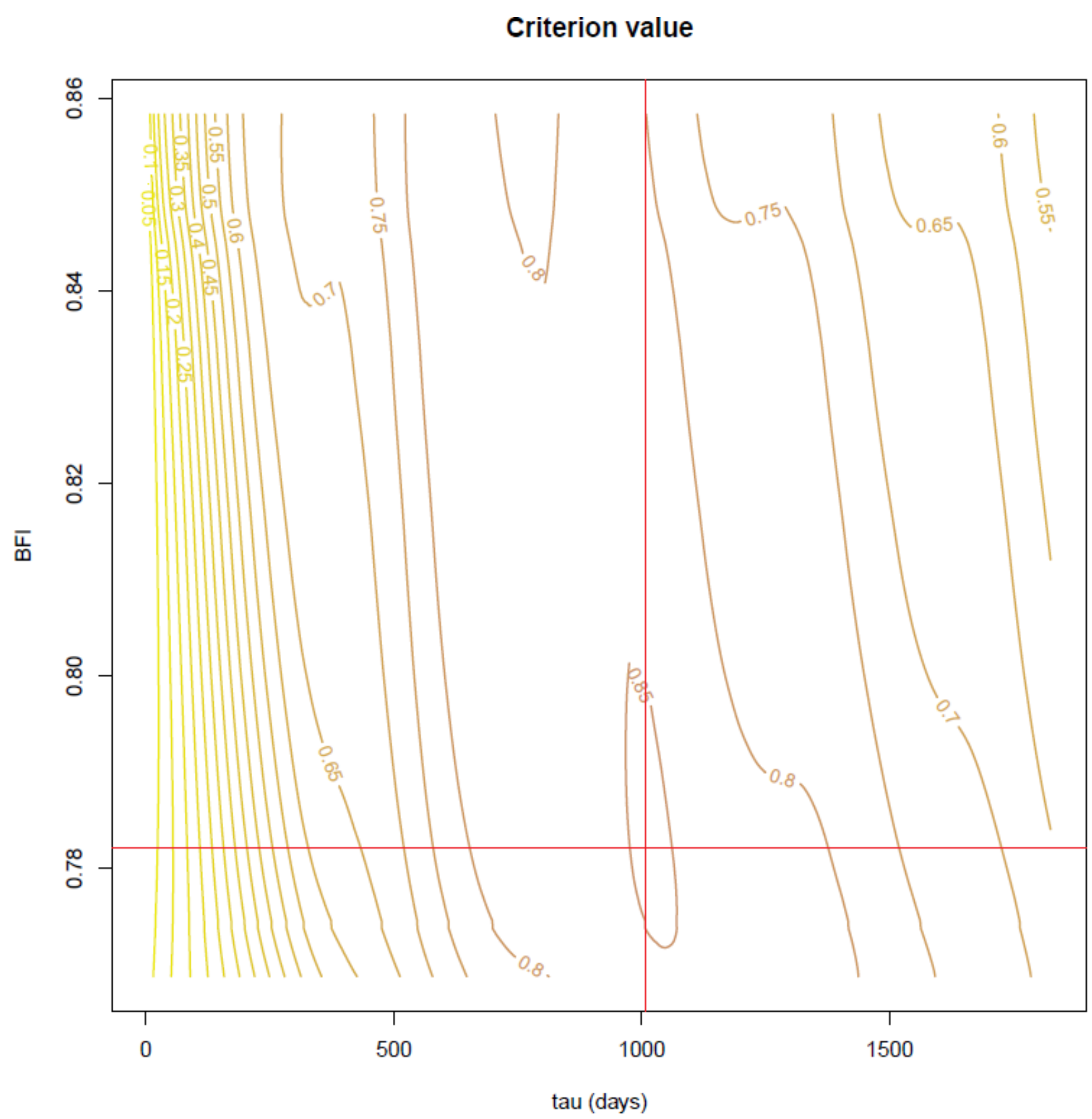

**Figure 4.** Surface plot of optimization criterion for the Petit Thérain river in Saint-Omer-en-Chaussée. Red cross indicates the numerical optimum.

with other local maxima. In agreement with geologic and climatic configurations, the optimized BFI (0.11) and $\tau$ (170 days) are low; the optimal value of $S$ is 4.42 m. The Petit Thérain river in Saint-Omer-en-Chaussée is heavily influenced by the large chalk aquifer in the Paris basin: its BFI and response time should be high. The criterion optimization is more equivocal here: there is a large plateau in the middle of the criterion map – around a correlation of 0.8 – which gives significantly different values of BFI – from 0.78 to 0.86 – and $\tau$ – from 700 to 1,400 days. However, a hill in the middle of the shelf indicates the optimal point, where the correlation is 0.86, is at BFI $=$ 0.783 and $\tau = 1{,}015$ days. Pseudo-periodical oscillations can be seen in parallel to the $\tau$ axis; they are consequences of the annual hydroclimatic cycle.

### 4.2 Hydrograph separations obtained

Figures 5, 6 and 7 show three examples of hydrograph separations obtained among the test set of catchments, with the respective BFI values of 0.11, 0.30 and 0.78. Figure 5 shows the example for the Vair river in Soulosse-sous-Saint-Élophe, a small catchment in the Meuse basin, with a small BFI. It lies on poorly permeable sandstone, marl and clay, with no connection to a

large aquifer. During dry periods, baseflow is almost determined by a local minimum on a sliding window with a length of 1 year, whereas it rises a little during wet periods, as seen during the 2000–2001 winter period.

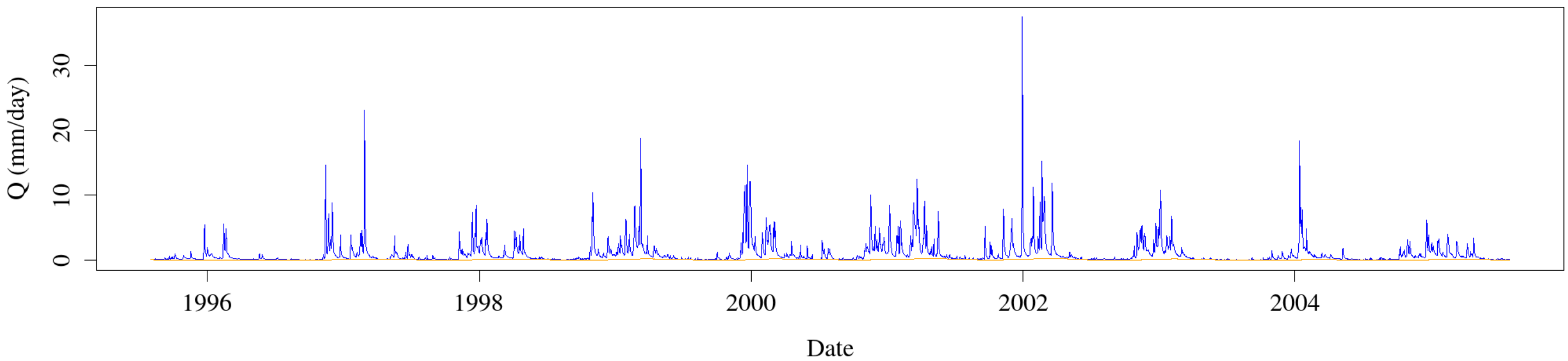

**Figure 5.** Hydrograph separation of the Vair river in Soulosse-sous-Saint-Élophe – 1995–2005.

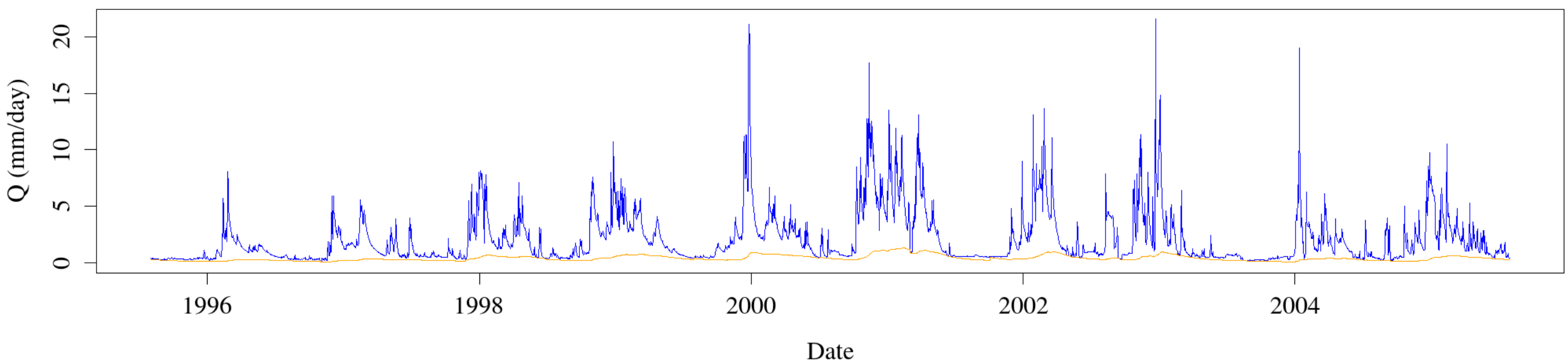

**Figure 6.** Hydrograph separation of the Virène river in Vire-Normandie – 1995–2005.

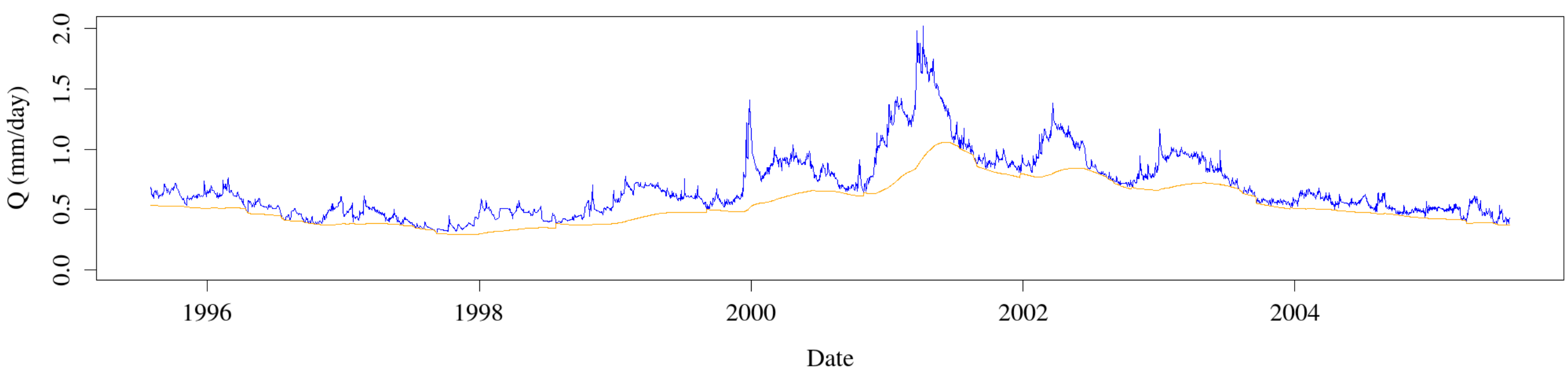

**Figure 7.** Hydrograph separation of the Petit Thérain river in Saint-Omer-en-Chaussée – 1995–2005.

Figure 6 displays a medium BFI case: the Virène river in Vire-Normandie is a granodioritic catchment located in the north-eastern part of the Armorican Massif, with a moderate aquifer influence. Baseflow fits the total flow during low-flow periods and rises during high flows, with a jagged shape made of asymmetric peaks: decreasing slopes fit the total flow – thanks to the

update procedure – whereas increasing slopes are gentler, driven by the smoothing effect of the reservoir. This creates a delay of baseflow with respect to total flow that resembles an aquifer's behaviour, with an outflow much higher just after a flood event than before. This baseflow recession resembles what would be obtained with a graphical separation method.

Figure 7 corresponds to a high BFI case: the majority of flow is identified as baseflow by the model. The Petit Thérain river in Saint-Omer-en-Chaussée is a small catchment fed by the well-known chalk aquifer of the Paris basin. Therefore, it is not surprising to find a BFI close to 1. An ensemble view to the hydrograph shows that long oscillations seem to influence total flow, with a period greater than one decade; it is well taken into account by baseflow separation.

Among the dataset of 1,664 catchments, optimisation of parameter $S$ failed for three catchments onl;: there, the correlation criterion kept increasing with the value of $S$, without the possibility of reaching an optimal value. The failing catchments are small basins where most of the water comes from glaciers in the Alps: effective rainfall – as computed by the Turc–Mezentsev formula – is thus not really correlated with baseflow and the quadratic reservoir fails to reproduce the particular memory effect of a glacier.

### 4.3 Synthesis and geological interpretation

**Table 3.** Summary of the results from the set of 1,664 catchments

| | BFI | $\tau$ (days) | Criterion value | S (m) |
|---|---|---|---|---|
| *Minimum* | 0.005 | 5 | 0.67 | 0.001 |
| *First quartile* | 0.103 | 165 | 0.74 | 1.98 |
| *Median* | 0.164 | 180 | 0.80 | 4.16 |
| *Mean* | 0.199 | 209 | 0.82 | 12.45 |
| *Third quartile* | 0.250 | 200 | 0.92 | 81.0 |
| *Maximum* | 0.896 | 1790 | 0.95 | 1,000 |

Table 3 presents a summary of the parameters identified by calibrating the separation algorithm on the 1,664 catchments. First, the criterion values obtained are quite high for Pearson correlations, with all values greater than $0.66$ and half of the catchments above $0.80$; baseflow is quite well-correlated with cumulated effective rainfall. Values of the reservoir parameter $S$ are huge, if we compare them with annual rainfall or with usual reservoir capacities for commonly used hydrological models; it is thus difficult to give them a physical meaning. The range of BFI obtained is large, between $0.01$ and $0.90$, with a median of $0.16$. The chosen catchments represent a wide variety of situations that can be taken into account by the separation algorithm. Finally, $\tau$ also covers a wide range of values, between $5$ and $1,790$ days with half of the catchments between $165$ and $200$ days. Concerning correlations between these results, we found that no clear relationship can be highlighted between themselves and with hydroclimatic values given in table 2, except for a positive Pearson correlation between $BFI$ and $\tau$ of $0.42$: we found no solution to reduce the number of parameters by inferring one from the other.
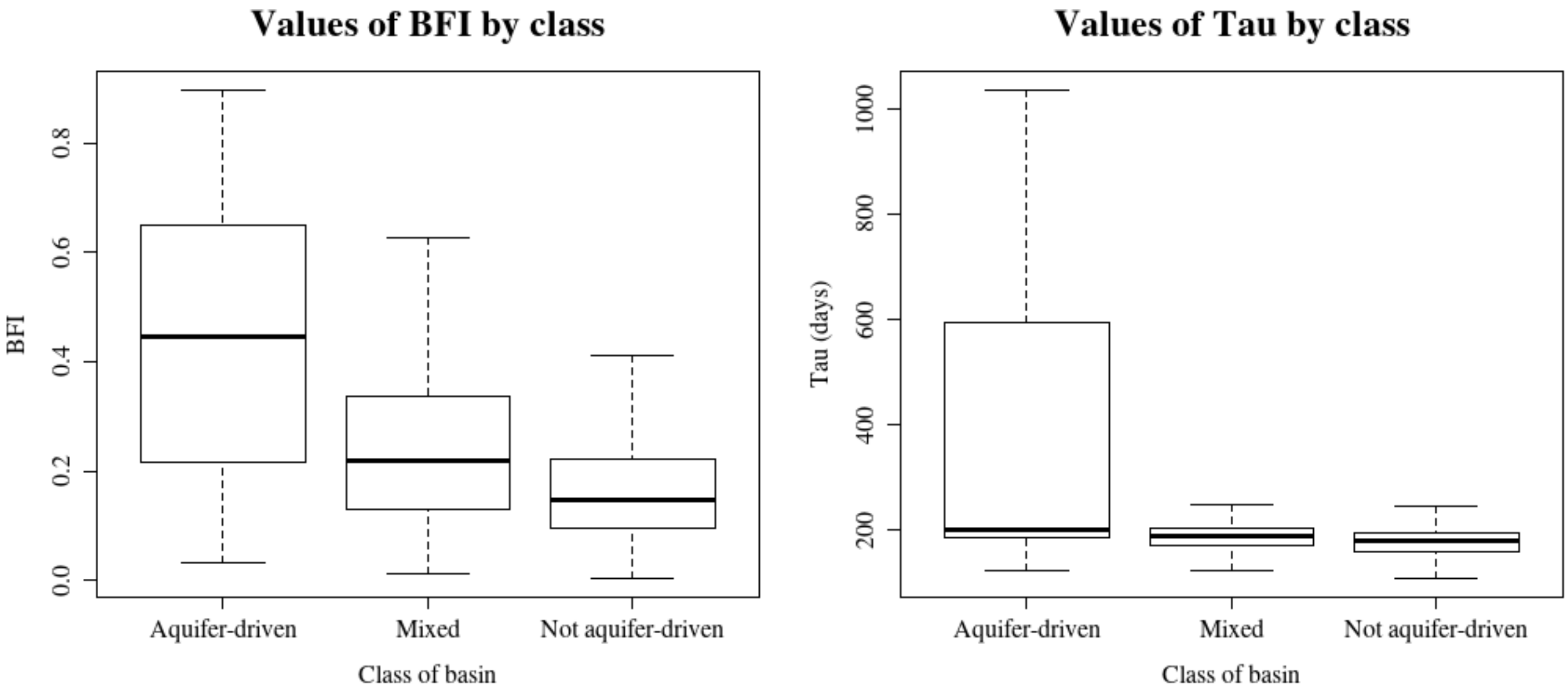

**Figure 8.** Boxplots of BFI values and $\tau$ according to the hydrogeological class of basins, i.e., whether they are likely to be under the influence of a significant aquifer. Central box accounts for the interval between first and third quartile; the segment in the middle is the median; whiskers represent 95 % confidence interval.

Figure 8 shows the distribution of BFI and $\tau$ values for each hydrogeological class of basin. The mean BFI is significantly larger for *aquifer-driven* basins than for other classes, but the ranges of BFI are not significantly separated: even *aquifer-driven* catchments can have low BFIs. However, those that are *not aquifer-driven* are limited to BFI values under $0.4$. All this agrees with the fact that baseflow is highly connected to the outflow of aquifers, whose contribution to total flow is much higher in catchments connected to large underground reservoirs. There are also catchments whose geological configuration suggests an *aquifer-driven* regime, whereas the BFI found is low; most of them are located above the Vosges sandstone aquifer which, although a crucial resource for water supply in this region, is not directly connected to piedmont rivers. As far as $\tau$ is concerned, most catchments are around 200 days, with a larger mean value for aquifer-driven catchments. It is notable that *mixed* or *not aquifer-driven* basins are limited to values of $\tau$ less than 300 days, whereas the range of *aquifer-driven* catchments' values extends to more than 1,000 days: this confirms the hypothesis that long memory is caused by the influence of an aquifer – even if the presence of the latter does not guarantee a long-memory behaviour.

Figure 9 shows geographic variations of the values of $\tau$ and BFI obtained. BFI values are coherent with the geological context: granitic regions like the Armorican massif or Massif Central have low values of BFI; high values are observed in the Alsace plain, in the sandy Landes of Gascony and particularly in the chalky zones of the Paris basin and the Somme basin. The rivers of these regions are known to be particularly under the influence of major aquifers that bear the memory of climatic events. The south-east area of France has medium BFI values, which is surprising since most of this region is composed of mountainous areas without any large aquifer; however, particularly smooths hydrographs can be explained by the presence of

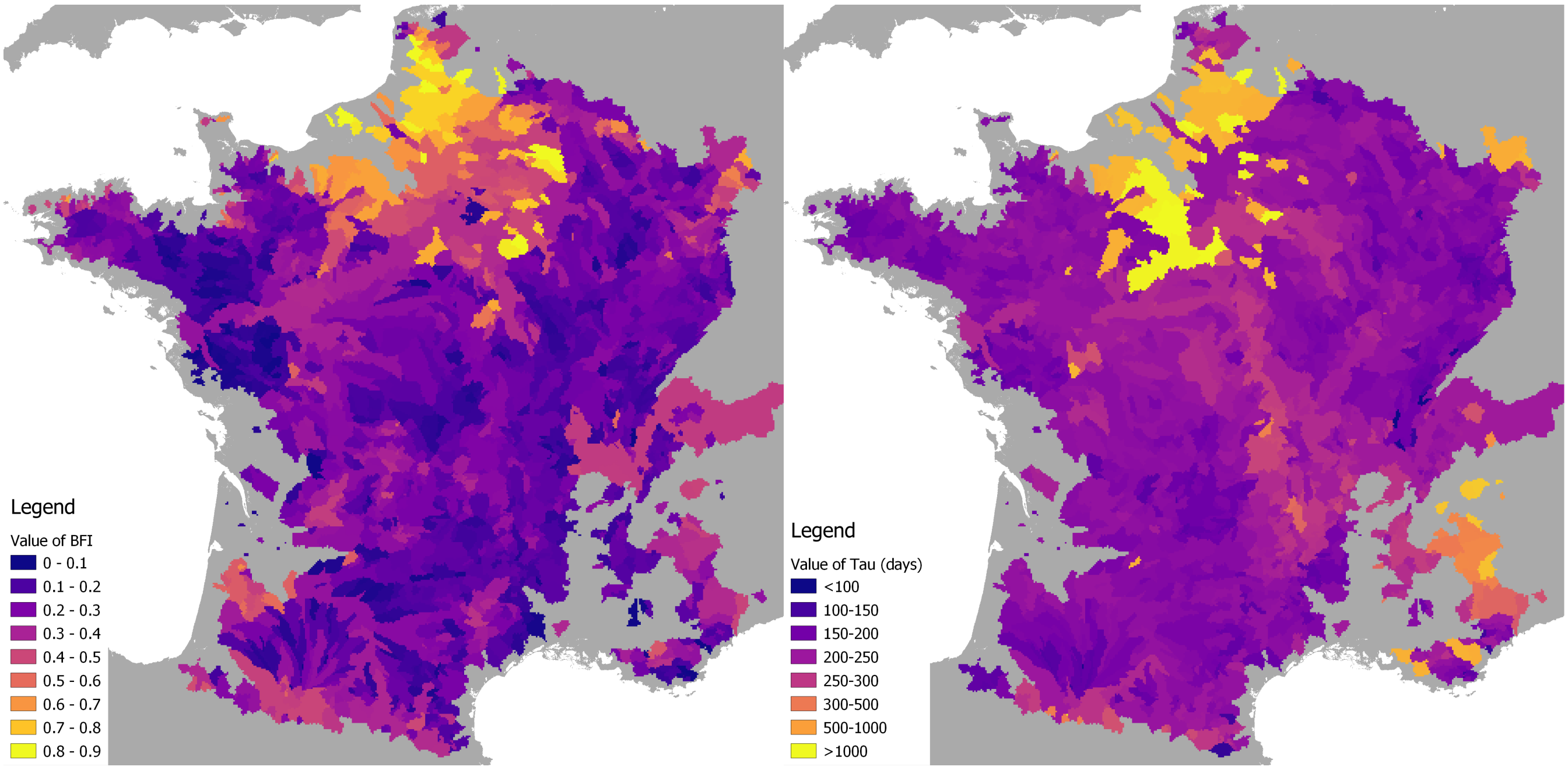

**Figure 9.** Maps of resulting values of parameters on the catchment dataset. Left: BFI; right: $\tau$.

many hydroelectric dams and natural lakes – the first of which is the Leman lake on the Rhone river – in this region, which have a significant influence on streamflow. Values of $\tau$ are more difficult to interpret: even though the chalk aquifer is clearly visible around the Paris region, high values of $\tau$ can be observed in the Alps, too, and medium values in the Massif Central. Here again, it can be explained by the presence of natural and artificial lakes that bear memory of past hydroclimatic events.

### 4.4 Split-sample test as a stability assessment

In order to evaluate the stability of the proposed algorithm and of its calibration procedure, a split-sample test was performed. The time extent of the time series of the dataset – 1958-08-01 to 2018-07-31 – was divided into two equal parts $P_1$ (1958-08-01 to 1988-07-31) and $P_2$ (1988-08-01 to 2018-07-31). The algorithm was then calibrated separately for each basin on these two periods and the values obtained for BFI and $\tau$ were then compared.

Figure 10 shows the graphical comparison of BFI and $\tau$ values obtained; table 4 displays two ways of statistical assessment that were performed after the split sample-test: correlation between values of BFI and $\tau$ and non-parametric Kolmogorov–Smirnov test (Smirnov (1939)), in order to check the equality of distributions, i.e., the bias between series of values.The BFI shows a good graphical consistency, with high Pearson correlation values – higher than 0.91 – but the p-values of the Kolmogorov–Smirnov test are under 0.05 for $P_1$; there is a constant bias between the BFI values computed for $P_1$ and those computed for the whole period. The scatterplot shows a good correlation between the BFI values computed for $P_1$ and $P_2$. As for $\tau$, it is less consistent than the BFI: the scatterplot shows some outliers that are far from the diagonal, which explains

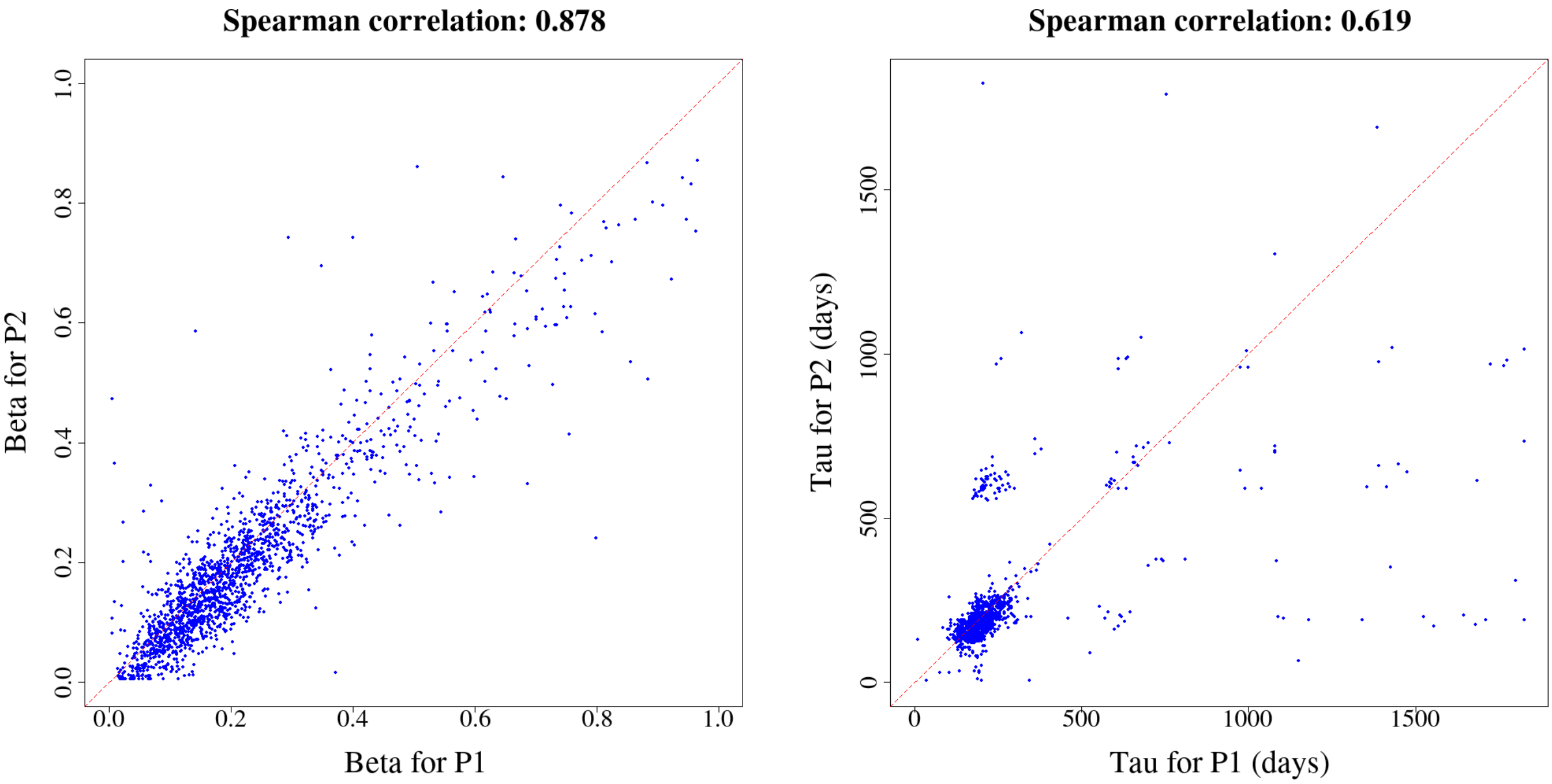

**Figure 10.** Results of split-sample test for BFI – on the left – and $\tau$ – on the right. Values obtained for $P_1$ and $P_2$ are compared.

**Table 4.** Statistical assessment of split-sample test through Pearson correlation and Kolmogorov–Smirnov (KS) non-parametric test.

| | BFI | | | | | | $\tau$ | | | | | |
|---|---|---|---|---|---|---|---|---|---|---|---|---|
| | **Pearson correlation** | | | **KS test p-value** | | | **Pearson correlation** | | | **KS test p-value** | | |
| | $P_1$ | $P_2$ | $P_{tot}$ | $P_1$ | $P_2$ | $P_{tot}$ | $P_1$ | $P_2$ | $P_{tot}$ | $P_1$ | $P_2$ | $P_{tot}$ |
| $P_1$ | 1 | 0.91 | 0.94 | 1 | <0.01 | <0.01 | 1 | 0.53 | 0.61 | 1 | <0.01 | <0.01 |
| $P_2$ | 0.91 | 1 | 0.98 | <0.01 | 1 | 0.42 | 0.53 | 1 | 0.84 | <0.01 | 1 | 0.28 |
| $P_{tot}$ | 0.94 | 0.98 | 1 | <0.01 | 0.42 | 1 | 0.61 | 0.84 | 1 | <0.01 | 0.28 | 1 |

the lower Pearson correlation values, between 0.53 and 0.84. Here again, values computed for $P_1$ are further away from those computed for the whole period than those computed for $P_2$. Despite these imperfections, this split-sample test can be regarded as successful, especially for a general-purpose analysis tool on a large set of catchments.

### 4.5 Comparison with a reference method

The BFI obtained was compared with the one resulting from a simple graphical method, based on local minima on a sliding interval of 5 days. It is detailed in Gustard and Demuth (2008). Figure 11 shows the scatterplot of the two computed BFIs. Gustard's method gives higher indexes than the conceptual filtering algorithm, but the trend remains, which is confirmed by the high Spearman correlation – 0.819. A high-BFI catchment according to Gustard's method will also get a high BFI from the

conceptual filtering and so it is for low values too. The difference between the results of the two methods can be explained by the time scales: the width of the sliding minimum window in Gustard's method is 5 days, whereas we get much larger values of $\tau$. Graphical quickflow is much quicker than the one resulting from the conceptual algorithm.

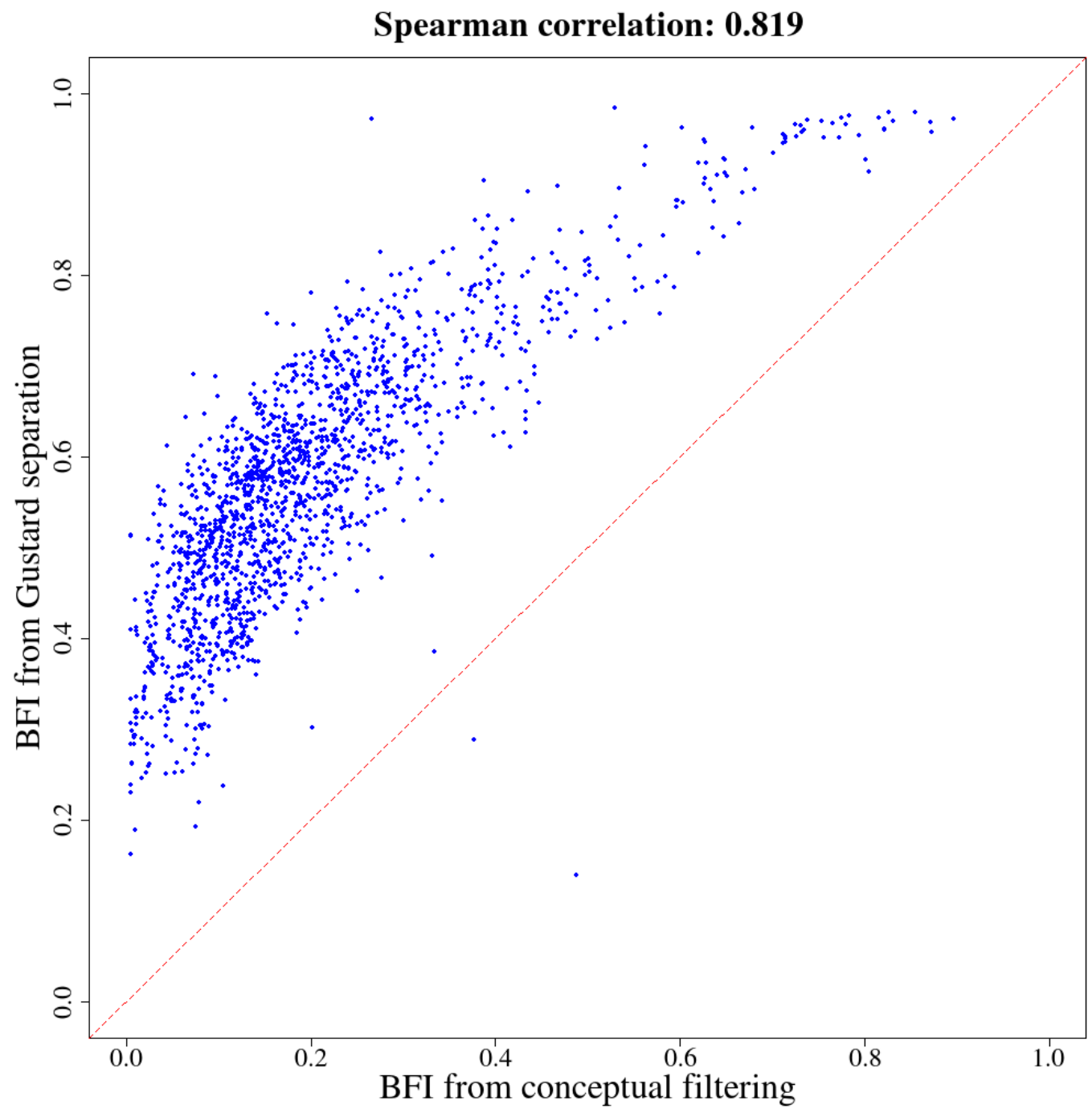

**Figure 11.** Comparison between BFI obtained with the conceptual algorithm and a reference graphical method.

## 5 Conclusion

### 5.1 Synthesis

The hydrograph separation algorithm and its calibration procedure presented in this work gave credible results for a large set of test catchments. The algorithm has suitable characteristics for such a model: (i) frugality – only two parameters to calibrate; (ii) objectivity – the procedure is not supposed to require any intervention or interpretation from the user; (iii) generalization ability – the procedure succeeded on the whole dataset, with a unique value found for the reservoir parameter; (iv) repeatability – as assessed by the split-sample test performed on the test dataset. The analysis of the results in the light of the geological characteristics of catchments shows that the model is able to acknowledge the importance of groundwater flow in total flow generation without prior knowledge about the studied catchment, which is a significant step forward with respect to other non-tracer-based baseflow separation methods.
A conceptual and empirical baseflow separation method is not intended to replace precise estimation of the elements of water balance at the scale of a given catchment, especially when infield physical data are available; neither can it precisely highlight particular hydrological processes occurring at the catchment or under-catchment scale, such as karstic non-linear transfers or human regulation of rivers. Yet, in the absence of infield data, i.e., outside of specially instrumented catchments, conceptual baseflow separation is a useful tool for obtaining meaningful insights about the role of groundwater and aquifers inside a catchment. Even if the present method has limitations – which we summarize below – it could be applied as an objective, general-purpose, automated analysis tool for information on a large set of flow series. Beyond the inherent characteristics of the presented method, the results show that the hydrogeological processes of a catchment are deeply engraved in streamflow.

### 5.2 Limitations

We used an optimization procedure set, which searches for values of $\tau$ less than 5 years. It showed great convergence ability, i.e., an optimal point was found for every catchment of the test dataset. Yet, a mere glance at the Petit Thérain river hydrograph shows that its response time may be greater than 5 years, with long dynamics at the scale of climatic cycles. All this highlights the fact that the simplistic distinction between *baseflow* and *quickflow* hides the diversity of time scales and travel times among processes that influence the total flow of a river.

The split-sample test is a way to check temporal consistency of the baseflow separation method; the one performed on the test dataset can be considered as successful, at least concerning BFI results. Nevertheless, the spatial coherence of the method was not checked; it could be tested, for instance, with propagation of baseflow from sub-basins of a larger catchment, in order to test whether the sum of tributary baseflows makes up total baseflow. Our optimization procedure does not take into account regional aspects while calibrating the model, although it could be a way of introducing objective prior knowledge in the method. Finally, the generalization capacity of the method has only been demonstrated in mainland France hydroclimatic conditions: even if the spatial extent of the present work has a substantial diversity of climates, further study with a wider range of climates and geologies is needed to validate the method.

### 5.3 Implementation

An R package named *baseflow* is provided to ensure the reproducibility of our results. It implements the algorithm presented in this paper with a high-performance computing core written in Rust. The example data taken from airGR are linked in the examples. The package is being released on CRAN (Comprehensive R Archive Network) and its documentation is available as supplementary material.

*Competing interests.* The authors declare that there are no competing interests.

Hydrology and Earth System Sciences
EGU
Discussions

*Acknowledgements.* Data for this work were provided by Météo France (climatic data) and the hydrometric services of the French government (flow data). We would like to thank Pierre NICOLLE and Benoît GENOT for the database creation and maintenance and José TUNQUI NEIRA for his help in the literature review. We would also like to extend our warmest thanks to Charles PERRIN and Lionel BERTHET for their remarks and suggestions during the writing process. Last but not least, computing codes could not not have been developed without the precious expertise and availability of Olivier DELAIGUE (Irstea Antony) and Denis MERIGOUX (INRIA Paris).

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
