# Peer review of "Hydrograph separation: an impartial parametrisation for an imperfect method"

_Hydrology and Earth System Sciences, 2019_

## Short Comment (SC1) · 17 Oct 2019

A quick question: a root transformed hydrograph

To my pleasantly surprise, the linear reservoir (Page 7, Line 1) is not the focus of the analysis, but the quadratic reservoir (Page 7, Line 2) is.

For Petit Thérain river (Figure 7), can the authors please share with us the annual hydrographs with the total measured streamflow $Q(t)$ and the computed baseflow $R(t)$

replotted in a negative reciprocal of the root (-RoR) or negative inverse square root (NISR) scale, $-1/\sqrt{Q}$?

For a quadratic reservoir or storage, the NISR transform linearizes the recession limbs for regression analysis, and displays as well the transformed data in a visually consistent frame for comparison with the logarithmic transform (See Santos et al., 2018, and therein SC2, SC5).

References

Santos, L., Thirel, G., and Perrin, C.: Technical note: Pitfalls in using log-transformed flows within the KGE criterion, Hydrol. Earth Syst. Sci., 22, 4583-4591, https://doi.org/10.5194/hess-22-4583-2018, 2018.

---

## Short Comment (SC2) · 21 Oct 2019

The aim of providing "an unarbitrary, impartial, repeatable parameterization that could be used as a general-purpose study tool over large catchment sets" is commendable but cannot escape the issues that have dogged the numerous methods in the past (in my own review of hydrograph separation in Beven, 1991, the section on choosing a separation method said simply "Don't").

In particular, it is surely poor hydrological practice to make any process interpretation of an arbitrary separation as the authors do here throughout – essentially treating quick-flow and surface flow as equivalent (though accepting that the latter might include some

interflow). But the differences between the type of separation presented here and the tracer separations between event and pre-event water tell us that this equivalence can be totally wrong (surely Figures 5,6,7 cannot support this equivalence?). Quickflow in many catchments can be made up of a large proportion of pre-event water (with some indication that this proportion might decrease with increasing event magnitude; sequences of events etc). This is a reflection of the differences between velocities and celerities in catchment responses (e.g. Beven, 2012; McDonnell and Beven, 2014).

But, this process interpretation is not actually necessary to the authors aims. They could simply present the method as differentiating between fast and slow responses – effectively as a way of estimating BFI. A consistent and standardised approach for regionalisation purposes could still be useful (e.g. Figure 9). The results would be different from another chosen method (as they show in Figure 11, so sometimes really quite different) but that reflects the arbitrariness of baseflow separation methods that they discuss in the introduction.

It also begs the question, however, of why the dividing line between fast and slow should be chosen in this way – why are there not intermediate responses that might be appropriate in some catchments but which are forced to be split between fast and slow by the method proposed here? This would almost certainly be reflected in insensitivity in the fitting surface plots for some catchments (such as Figure 3 & 4). The DBM methodology of Young (2013 and references therein) could, for example, provide one way of providing a justification for the number of components by identifying the order of the transfer function, but this could well vary for different catchments (including cases where it is not necessary to invoke a slower 2nd order or 3rd order components; the discharge is adequately described by a first order component).

So it seems to me that baseflow separation still remains a rather desperate technique with somewhat arbitrary results depending on whatever storage or filtering function is chosen. Interpretation in terms of any process interpretation should definitely be avoided. As demonstrated in the paper it can certainly provide estimates of a BFI –

but it is not the BFI since it depends on a particular set of assumptions (the quadratic storage in this case) that might not apply everywhere (or actually anywhere).

References

Beven, K.J. (1991), Hydrograph Separation?, Proc.BHS Third National Hydrology Symposium, Institute of Hydrology, Wallingford, 3.1-3.8.

Beven, K.J. (2012) Rainfall-Runoff Modelling: The Primer, Wiley-Blackwell: Chichester

McDonnell, J J and Beven, K J, 2014, Debates—The future of hydrological sciences: A (common) path forward? A call to action aimed at understanding velocities, celerities, and residence time distributions of the headwater hydrograph, Water Resour. Res., 50, doi:10.1002/2013WR015141.

Young, P. C., 2013, Hypothetico-inductive data-based mechanistic modeling of hydrological systems, Water Resour. Res., 49, doi:10.1002/wrcr.20068.

---

## Author Comment (AC1) · 22 Oct 2019

We would like to thank Dr J. Ding for reading our manuscript and for his comment. The transformation he suggests is very interesting to illustrate the way streamflow is filtered in the algorithm presented in the manuscript.

The negative inverse square root (NISR) transformation is a useful tool to highlight the behaviour of a quadratic reservoir during recessions. Indeed, if the input of the quadratic reservoir is set to zero – i.e. when the fraction of streamflow $\beta Q(t)$, the input of the reservoir, is negligible compared to the baseflow $R(t)$, the output of the reservoir – the system of equations (1) and (2) – page 6 – becomes:

[Figure]

$$\frac{dV}{dt} = -\frac{V^2}{S\Delta t} \tag{1}$$

$$R(t) = \frac{V^2}{S\Delta t} \tag{2}$$

If we integrate these equations given an initial condition $V(t_0) = V_0$, we get the following NISR-transformed recession curve:

$$\frac{1}{\sqrt{R(t)}} = \frac{\sqrt{S\Delta t}}{V_0} + \frac{t - t_0}{\sqrt{S\Delta t}} \tag{3}$$

This curve is a straight line: therefore, during low-flow periods, we would expect the baseflow curve computed by the filtering algorithm presented in the manuscript to be close to a line segment. Figures 1, 2 and 3 show the three separated hydrographs shown in page 16 of the manuscript, with a NISR transformation applied to measured streamflow and baseflow. In figure 1, baseflow index is low and thus, during low-flow periods, the inflow of the quadratic reservoir is negligible compared to computed baseflow. We can observe that several parts of the transformed baseflow curve are line segments; this happens during low-flow periods. This behaviour can also be observed, to a lesser extent, in figure 2, for instance during the 1996–1997 drought. In figure 3, the condition to get a straight recession curve like in equation 3 are never met, because baseflow index is too high and therefore, the input of the reservoir is never negligible compared to its output. Yet, the NISR transformation points out the behaviour of the separation algorithm during low-flows and the role of updates is clearly visible. It also highlights the smoothing role of the filter.

In a nutshell, we consider the negative inverse square root transformation as a good tool to show the behaviour of the hydrograph separation algorithm on a wide variety

of catchments. We would like to thank Dr. J. Ding for his suggestion and we will add comments as a appendix. Obviously, we will acknowledge his contribution in the text.

[Figure]

[Figure]

**Fig. 1.** NISR-transformed separated hydrograph of Vair river in Soulousse-sous-Saint-Élophe

[Figure]

**Fig. 2.** NISR-transformed separated hydrograph of Virène river in Vire-Normandie

[Figure]

**Fig. 3.** NISR-transformed separated hydrograph of the Petit Thérain river in Saint-Omer-en-Chaussée

---

## Short Comment (SC3) · 24 Oct 2019

A first approximation of a baseflow hydrograph

I appreciate the prompt response to my request for an NISR-transformed hydrograph plot for the Petit Thérain river. The two additional plots the authors provided for Vair and Virène rivers and their positive comment are much appreciated.

Their new set of the recession equations (2) and (3) can be combined and the sign

changed to an explicit form of the NISR-transformed recession curve as follows:

$$-\frac{1}{\sqrt{R(t)}} = -\frac{1}{\sqrt{R(t_0)}} - \frac{1}{\sqrt{S\Delta T}}(t - t_0),$$

For the Vair river (new Figure 1) having a lowest baseflow index (BFI) of 0.11 among the three rivers, the resemblance among the annual cyclical variations is stunning.

The measured falling limb ordinates, say, $Q_-(t) = Q(t)$ whenever $-1/\sqrt{Q(t)} < -1/\sqrt{Q(t-1)}$, are indicative or first approximation of a baseflow hydrograph.

For the Vair river at least, can the authors please show all these falling limbs $Q_-(t)$, the computed baseflows $R(t)$, and the differences between the two, $Q_-(t) - R(t)$?

―――――――――――――――――――

---

## Short Comment (SC4) · 25 Oct 2019

Clearly, prohibition has not worked since I wrote my summary almost 20 years ago!! But that does not mean that the definition of base flow is any clearer now than then. And that is not just a matter of single storm separations versus long term separations, since environmental isotope data can be used over long time series to infer velocity-related (and process based) travel time distributions. These generally reveal no clear separation between slow and fast, but a full spectrum of travel times. That will also be true of the celerities that control the rising and falling limbs of the hydrographic.

As I said in my comment I fully understand the aims of the authors in presenting their
methodology but they should explicitly recognise that they are producing AN estimate of base flow (which, depending on their particular assumptions, cannot be considered definitive as THE estimate of base flow). Applied in the way they have done so, consistently across many basins, it might a useful tool - but please remove any process interpretation in revising the paper.

———————————————

---

## Author Comment (AC2) · 25 Oct 2019

We would like to thank Pr. Keith BEVEN for reading our manuscript and for his short comment. It will help us to state more explicitly our intentions with the new hydrograph separation method presented in the manuscript.

[Figure]

**Prohibition does not work**

Pr. Beven's review on hydrograph separation() recommands not to use any of the presented methods, with a short statement: "Don't". We understand that hydrograph separation is anything else but a good hydrological modelling approach; in particular, it is not supposed to have any predictive capacity. However, the interwar period in the USA has shown that prohibition does not work, for alcohol but also for hydrograph separation. We think that telling hydrologists not to use this age-old simple hydrograph processing technique will not prevent them to use subjective and partial graphical or empirical methods, and therefore, there is an interest in developing an unarbitrary, impartial and repeatable parametrization of a simple conceptual method.

**Do we all agree on what baseflow is?**

The definition of baseflow itself is a problem. As underlined in Pr. Beven's comment, any process-based definition of baseflow leads to a slippery slope: a quick response to a rainfall event is not necessarily composed of recent water and it can even be mostly old-water, mobilised from sub-surface and ground(). It is the difference between celerity – or phase speed – which is the propagation speed of the perturbation caused by the rainfall event, and velocity – or group speed – which is the speed of the molecules of water themselves(). The baseflow that we would like to identify with the method presented in the manuscript is composed of slowly responding components of streamflow, i.e. perturbations propagated with a low celerity and much dispersion, which can be composed of recent or old water. We will improve the definition of baseflow given in the manuscript, to detach ourselves from the simplistic process-based interpretation of hydrograph separation.

**Events are not relevant**

The discussion about the age of water and the distinction between velocity and celerity is relevant at the scale of a single event; for instance, to study the precise components and their dynamics of an individual flood peak, the method presented in the manuscript has almost no interest. It is intended to analyse long time series of streamflow, to identify the components that bear long-term memory of past hydroclimatic events. For the sake of simplicity, total streamflow is divided into only two components: the quick one and the slow one. The computation of BFI yields a global indicator to highlight *baseflow-dominated*() catchments and in our national analysis, it is quite well correlated to the geological context: aquifer-driven catchments have the highest BFIs. This purpose of the method will be explained more explicitly in the revised version of the manuscript.

**References**

K. Beven. Hydrograph separation? In *Proc.BHS Third National Hydrology Symposium*, pages 3.2–3.8. Institute of hydrology, 1991.

James W. Kirchner. A double paradox in catchment hydrology and geochemistry. *Hydrological Processes*, 17(4):871–874, 2003.

Jeffrey J. McDonnell and Keith Beven. Debatesâ˘ATˇthe future of hydrological sciences: A (common) path forward? A call to action aimed at understanding velocities, celerities and residence time distributions of the headwater hydrograph. *Water Resources Research*, 50(6):5342–5350, 2014.

Marios Sophocleous. Interactions between groundwater and surface water: the state of the science. *Hydrogeology Journal*, 10(1):52–67, Feb 2002.

---

## Author Comment (AC4) · 5 Nov 2019

Thank you for this helpful exchange. We will incorporate your remarks in the revised version of the article, through a clearer definition of baseflow and our purpose with the computation of one of its estimate. In particular, process-based interpretation will be avoided in the revised manuscript.

---

## Author Comment (AC5) · 26 Nov 2019

We would like to thank Dr J. Ding for his comment. The representation he suggests illustrates how the algorithm behaves during recessions.

In the 3 example hydrographs shown in the article, we isolated recessions – i.e. periods of time in which streamflow is decreasing – longer than 15 days and for each event, we plotted NISR-transformed streamflow, NISR-transformed baseflow and the difference between the two. The results are shown on figures 1, 2 and 3.

The method presented in the article is not intended to study particular recession events,

but to be applied on a whole hydrograph of several hydrological years. That is why it is difficult to analyse the events graphs below. However, we can see two phenomenons:

- the baseflow peak is sometimes delayed with respect to the streamflow peak: it represents the delayed effect of the slow component of streamflow, with respect to the quick component;

- in most events, streamflow converges towards baseflow at the end of the recession. At least, the difference is decreasing during recession, since baseflow is closer to streamflow during low-flows than during flood peaks.

This analysis is consistent with what we intend to do with the hydrograph separation method presented in the article.
* * *
[Figure]

**Fig. 1.** NISR-transformed separated recessions of Vair river in Soulousse-sous-Saint-Élophe. In blue, NISR-transformed streamflow; in orange, NISR-transformed baseflow; in purple, the difference.

**Fig. 2.** NISR-transformed separated recessions of the Petit Thérain river in Saint-Omer-en-Chaussée. In blue, NISR-transformed streamflow; in orange, NISR-transformed baseflow; in purple, the difference

**Fig. 3.** NISR-transformed separated recessions of Virène river in Vire-Normandie. In blue, NISR-transformed streamflow; in orange, NISR-transformed baseflow; in purple, the difference

---

## Referee Comment (RC1) · Renata Romanowicz (Referee) · 8 Dec 2019

General comments:

The authors present a new method for hydrograph separation. As the title suggests, the method is not perfect but in my opinion it is a big step forward in the estimation of baseflow. Referring to the discussion on the HESSD web-page, even though a separation of baseflow from total flow is impossible, baseflow remains an important characteristics of river flow. The novelty of the approach introduced by the authors consists in relating the baseflow model parameters to catchment hydrogeological conditions. This is done by looking for a correlation between baseflow values and the time response

of a river basin expressed as a cumulative effective rainfall. In other words, the BFI model parameter, basin storage, is defined as that which gives the largest correlation between baseflow time series and cumulative effective rainfall. The underlying hypothesis is that those catchments which have larger baseflow indices due to recharge from the groundwater storage will have longer time responses. Based on the results from the 1664 French catchments, the authors provide evidence of a relationship between the geological characteristics of a catchment and the BFI values. Even though this hypothesis is interesting and gives consistent baseflow values, it is imperfect due to the lack of a clearly defined optimisation criterion. That suggests that maybe some more work towards the improvement of that criterion is required. The other point is the assumption of a linear relationship between total flow and groundwater recharge. It is a strong assumption and maybe some more discussion should be given on the catchment conditions when it can be fulfilled and when it is not likely to be kept.

Specific comments:

Pages 6 and 7: The notation is rather confusing -the authors mix the discrete notation with continuous.

Equation for Q at page 7, line 15 is not needed and is rather confusing, as I presume that 'tau' in that equation is not the same as 'tau' on page 8?

Page 7 lines below equation for Q –the explanation of the integration scheme is not very clear

Page 10, Algorithm 1: there is a mistake in the Ri substitution (should be beta*Qi) following eq. 1

Page 17: lines 21-23 – this is important information and could be more elaborated on and made clearer.

Page 20, Fig. 10, left panel axis should be corrected (BFI instead of Beta).

---

## Referee Comment (RC2) · Ian Cartwright (Referee) · 12 Dec 2019

Review of "Hydrograph separation: an impartial parametrization for an imperfect method"

Ian Cartwright

General Comments

This paper is the latest in a long series of efforts to make robust and automated or reproducible separations of baseflow. In doing so the authors may have improved the methodology and using catchment characteristics is desirable. The application

to a variety of catchments in France illustrates its capacity to provide regionalisation of baseflow estimates over a wide range of catchment types (rainfall, topography, geology etc).

The paper is generally understandable but in places the writing is idiomatic and not very precise. I do not think that papers need to be overly formal in their writing style but the final version should probably be a little less conversational in style.

My main concern with this paper is not the new methodology or its application but more fundamentally, what is it that we are calculating when we perform two-component hydrograph separations. As outlined in several publications (including Gonzales et al., 2009 [cited] and Cartwright et al. 2014: Hydrology and Earth System Sciences, 18, 15-30), baseflow estimates based on hydrographs (eg filters or graphical separations) are commonly higher than those based on chemical mass balance. Some studies have sought to "tune" filters using geochemistry (this is discussed in Section 2.4 and was employed by Gonzales); however, this may miss the point that the system involves more than two components.

River flow comprises surface runoff, groundwater inflows, and a range of intermediate stores of water (bank return flows, interflow, water stored in pools on the floodplain). These various stores probably contribute differently to river flow at different stages of the hydrological cycle. If we define baseflow (or slow flow) as all the delayed stores of water, then it may consist of any and all of these stores and not just be groundwater (i.e. the deeper waters contained in the aquifers). These intermediate stores of water are likely to have a geochemistry more similar to that of the surface runoff (especially if residence times are insufficient for significant geochemical reactions in the catchment to progress), which may explain the discordance between estimates of baseflow from chemical mass balance and hydrograph techniques. The paper needs to be more explicit on these points.

For example:

Section 2.5. "Streamflow in a river is seen as having two origins, groundwater – the sum of contributions of various aquifers – and surface water, made of surface runoff and subsurface interflow, i.e., water that does not stay too long underground. Surface water response to climatic events is much quicker than that of groundwater and this speed difference is time-coherent all along the hydrograph."

Section 3.1 "which underpins the concept of baseflow – can be represented by a conceptual reservoir, whose outflow will represent the groundwater contribution to streamflow".

Section 3.1.1. "What recharges the aquifer with water that will be baseflow afterwards?". The part of rainfall that does not contribute to surface runoff or to evaporation is generally named recharge, as it is thought to feed groundwater storage."

These examples implicitly assume that all delayed inputs are from groundwater. This is an oversimplification of the system and something that needs to be acknowledged and addressed. Presumably the time responses of these other reservoirs are shorter than for groundwater (which may be problematic). This is an important point as it does not matter how good the concepts and their applications are if the conceptualisation is not correct or well explained

Overall, I think the paper is an advance to this field but it might be more imperfect than they conceive. It certainly removes some of the arbitrary and more difficult parameterisation of other techniques. Attention to the points above and the possible other limitations (see below) would improve its impact.

Specific comments

Abstract

This section is too brief (it reads like a highlights section). Put some more detail in to emphasise what new understanding of hydrology you made and/or the limitations of the technique (see later)

Introduction (Section 1) and Hydrograph separation: a short review (Section 2)

These sections should be combined. The order is fine but Section 2 really just provides more background material, so is really part of the introduction

The literature is also rather dated; it is good to see the older classic discussions included but there have been more recent papers. Please add some more recent studies.

Section 3

The number of variables make reading this section difficult and there may be some duplication of parameters (I think that Tau is used for two different variables – I may be wrong here but check). I would suggest adding a Table of variables, which would help the readability.

Some of the explanations here are also not very clear to understand and should be read carefully for clarity

The text "For the sake of simplicity, a straightforward hypothesis is added: baseflow must be equal to total flow at least once in a hydrological year, when measured streamflow reaches its yearly minimum" glosses over a major issue. This may be a desirable simplification but is it correct? This is a common assumption for perennial streams in semi-arid areas with long dry periods but is it justifiable in the higher rainfall areas of France? Like several other assumptions in the paper, this rather bald assumption detracts from the study and needs more justification.

In a similar way "In this paper, we make the hypothesis that the best candidate for a proxy of recharge is a linear fraction of total flow itself. It is quite well-correlated with the behaviour of recharge given above. In addition, it is available without a further model or hypothesis." glosses over a major issue. Intuitively, I would have thought that recharge is not a linear function of flow as at high flows the ration of streamflow to baseflow is higher. More justification is also needed here.
With all these justifications, the text may become long. If so, the details of the mathematics could go in an appendix / supplement and the important conceptualisations / justifications in the main paper.

The response time (Tau) is a critical parameter in the context of this paper. You should define exactly what it is.

Section 4

Do the authors have sufficient temporal geochemistry data from any of the catchments to look at whether the baseflow separations can predict the geochemistry of the rivers. This might be something as simple as EC data. While the chemical mass balance approach to hydrograph separations is not foolproof, it does allow some checking to be made (the Gonzales et al., and Cartwright et al., papers amongst many other studies looked at this). It might go someway to answering the question as to how many stores of water there may be.

Looking at the geochemistry may also help in justifying the assumption that at low flows, the river is fed entirely by groundwater as at those times one would expect that the river geochemistry is very similar to that of groundwater in near-river shallow aquifers. This would be difficult for the larger rivers that could be fed by multiple aquifers but it may be possible for the smaller rivers.

Section 5

This section is a rather short summary of the findings. One aspect that needs addressing is what the limitations of this technique are. Most baseflow estimates based on hydrographs have some limitations around

1) Regulated rivers (i.e. those with dams or barrages on them)

2) Areas where groundwater abstraction results in disconnection

3) Rivers that receive inputs from canals or the like

4) Rivers where major pumping of water occurs

Is this the case here and can you outline practical limits where you think that this technique applies less well? Can you discuss the implications for applying this in practice?

―――――――――――――――――

---

## Author Comment (AC6) · 16 Dec 2019

We would like to thank Pr. Renata Romanowicz for reading our manuscript and for her careful and useful review. Here are our answers to the points raised by her remarks.

**Optimisation criterion**

As it is highlighted in Pr. Romanowicz's comment, the optimisation criterion that we use to calibrate the parameters of the hydrograph separation algorithm – Pearson correla-

tion between baseflow and cumulated effective rainfall – is not univocal enough to be a satisfying correlation criterion. Indeed, it is subject to pseudo-periodical oscillations caused by the annual hydroclimatic cycle; it is visible on figure 3, page 14. We chose to use Pearson correlation since it is quick to compute and easy to interpret; however, a more complex optimization criterion could be developed indeed to get more univocal results.

**Linear relationship between flow and recharge**

Pr Romanowicz underlined a major hypothesis of the hydrograph separation method presented in the manuscript: the water inflow of the quadratic reservoir is a linear fraction of daily measured streamflow. This is a very crude estimate of aquifer recharge, which is a far more complex process including water flow through banks of the river, soil water balance, vegetation, seasonality, etc. Solving the groundwater transmissivity equation in a theoretical framework of a shallow aquifer connected to a river shows that recharge in anything but a linear fraction of streamflow; and real configurations of river-aquifer interactions are even more intricate.

However, as highlighted by the exchange with Pr. Keith Beven, we do not claim to present a physically-based hydrograph separation method. For such a purpose, we would need an explicit recharge model, that would add more hypotheses and parameters: an elaborate production function in such an imperfect, but objective, algorithm would be as useful as a chocolate teapot. Therefore, the inflow function composed of a fixed linear fraction of daily streamflow can be regarded as a basic estimate of the quantity of water that the catchment remembers.

In the revised version of the article, we will add a clearer explanation about the inflow function of the reservoir.

[Figure]

**Equations in pages 6 and 7**

We use a different notation for the continuous and the discrete versions of a variable: $X(t)$ for the continuous one and $X_t$ for the discrete one. Integration of the continuous differential equations is made through a Eulerian explicit scheme and equation on page 7, line 15 is intended to explain how continuous variables are made from discrete measurements; in the revised version of the manuscript, we will replace it by a clearer plain-text explanation that will avoid confusion with other variables in the algorithm. We will also add a diagram to explain the integration scheme.

**Correlation between parameters**

Two parameters need to be calibrated in the algorithm presented in article. We tried to find a simple relationship between catchments' characteristics and parameters, in order to remove one degree of freedom in the optimization process; but we did not manage to find one. Since it is an unsuccessful point, it is not detailed in the manuscript; it is only mentioned at the end of page 17. In the revised version of the manuscript, we will give more details about correlations between parameters.

Algorithm 1 :

Pr. Romanowicz noticed that $\beta$ is missing at line 6, according to equation 1. It will be corrected in the revised version of manuscript.

Figure 10 :

Pr. Romanowicz noticed an error in the left panel y-axis label: it will be corrected as BFI.

---

## Author Comment (AC7) · 17 Dec 2019

Antoine Pelletier and Vazken Andréassian

antoine.pelletier@irstea.fr

We would like to thank Pr. Ian Cartwright for reading our manuscript and for his careful and useful review. Here are our answers to the points raised by his remarks.

**Hydrograph separation and hydrological processes**

The separation of streamflow in two components – the slow and the quick one, or more precisely the delayed and the non-delayed one – is artificial: there is generally a wider

range of hydrological processes at stake that have different response times to hydroclimatic events. Having said that, it is still possible to set up an arbitrary barrier between the *not too delayed* components of streamflow and the *delayed enough* components.

But attributing the flow component to an explicit physical process seems to be risky with a conceptual separation method like the one we present in the manuscript. Pr. Cartwright highlights that we define baseflow as the groundwater component of streamflow and quick flow as the surface runoff component. This distinction is improper and partial, we agree that it is an oversimplification of the system. Ideally, we should take into account various intermediate stores of water, for which we do not want to make hypotheses about their water residence time.

Anyhow, process interpretation of a conceptual separation method is hazardous practice – this was highlighted too by the interactive discussion with Pr. Beven. In the revised version of the article, after explaining the various types of water sources and their response behaviour, we will define baseflow as the sum of delayed streamflow components, whatever source they are from. The interest of this component is that it bears inter-annual memory of a catchment.

**Abstract**

We believe in short and catchy abstracts that are likely to be read entirely. However, we agree that some crucial elements are missing in the proposed version. We propose this new abstract, which is a little longer.

This paper presents a new method for hydrograph separation. It is well-known that all hydrological methods aiming at separating streamflow into baseflow – its slow or delayed component – and quickflow – its non-delayed component – present large imperfections, and we do not claim to provide

here a perfect solution. However, the method described here is at least (i) impartial in the determination of its two parameters (a quadratic reservoir capacity and a response time), (ii) coherent in time (as assessed by a split-sample test) and (iii) geologically coherent (an exhaustive validation on 1,664 French catchments shows a good match with what we know of France's hydrogeology). With these characteristics, the method can be used to perform a general assessment of hydroclimatic memory of catchments. Last, an R package is provided to ensure reproducibility of the results presented.

**Introduction and review section**

In the revised version of the article, we will merge the two sections. As underlined by Pr. Cartwright, some recent references are missing in the literature review. We propose to add several citations listed in the references section of this comment.

**Section 3**

Readability of the equations

As suggested by Pr. Cartwright, we will add a table of variables to help the readability. We will also replace the lengthy explanation of the integration scheme through an equation including Dirac functions on page 7 by a plain-text explanation.

Reservoir update

In the presented algorithm, there are two ways the level of the reservoir can be updated: downward, when the computed baseflow is greater than measured streamflow; upward, once a year, at the yearly minimum point of measured streamflow. The first one is understandable – baseflow cannot be greater than total streamflow – and the second one is just practical: as highlighted by Pr. Cartwright, in humid temperate French climate, the hypothesis that streamflow is only composed of baseflow at its yearly minimum is questionable, even though most recent French summers were dry enough to support this assumption. However, we need an upward update mechanism to compensate the downward one, since it allows the algorithm to be constrained to a water balance condition – that can be summed up by $\beta = BFI$. Removing the yearly upward update adds a degree of freedom for the model calibration and it causes significant difficulty to optimize the values of parameters. We will add this discussion in the revised version of the paper.

Recharge as linear fraction of streamflow

This issue was underlined by both referees, Pr. Romanowicz and Pr. Cartwright: the fact that the water inflow of the quadratic reservoir is a linear fraction of daily measured streamflow is a major hypothesis of the algorithm. This is a very crude estimate of aquifer/subsurface recharge, which is a far more complex process including water flow through banks of the river, soil water balance, vegetation, seasonality, etc. Solving the groundwater transmissivity equation in a theoretical framework of a shallow aquifer connected to a river shows that recharge in anything but a linear fraction of streamflow; and real configurations of river-aquifer interactions are even more intricate.

However, as highlighted by the exchange with Pr. Keith Beven, we do not claim to present a physically-based hydrograph separation method. For such a purpose, we

would need an explicit recharge model, that would add more hypotheses and parameters: an elaborate production function in such an imperfect, but objective, algorithm would be a disproportionate weapon. Therefore, the inflow function composed of a fixed linear fraction of daily streamflow can be regarded as a basic estimate of the quantity of water that bears the catchment memory.

In the revised version of the article, we will add a clearer explanation about the inflow function of the reservoir.

Tau parameter

The idea of the optimization criterion used to calibrate the parameters is the following: we try to correlate two estimates of the quantity of water available to bear the catchment memory. First, computed baseflow; second, medium-term or long-term cumulative effective rainfall – which we estimated through the Turc-Mezentsev formula. Parameter $\tau$ is the length of the cumulating period for effective rainfall, it is an estimate of the response time of the catchment. Unfortunately, it gave less consistent results than the computed values of baseflow index, due to more calibration difficulty.

**Section 4**

We agree with Pr. Cartwright that using geochemical data would be a less questionable way to calibrate our hydrograph separation process than the hydroclimatic criterion that we use. Unfortunately, we do not have available enough geochemical data in the French catchments of the sample to carry out a reliable study. We will mention this idea as a perspective.

**Section 5**

The technique presented in the manuscript is designed to be used on non-anthropised catchments and perennial streams. As highlighted by Pr. Cartwright, these limitations need to be more explicitly exposed in this section. When the value of streamflow is affected by human activity – pumping, regulation by dams, canals, etc. – the algorithm can be applied only if the anthropised fraction of streamflow is small enough. This is a classical check in rainfall-runoff modelling. The case of groundwater abstraction resulting in disconnection is different: it is still caused by human activity but it indirectly affects streamflow. In such a case, if the aquifer contribution to the river is zero; but in the algorithm, reaching an empty filtering reservoir could result in some artefacts. This is a clear practical limitation of the method.

**References**

Beven, K. (1991). Hydrograph separation? In *Proc.BHS Third National Hydrology Symposium*, pages 3.2–3.8. Institute of hydrology.

Cartwright, I., Gilfedder, B., and Hofmann, H. (2014). Contrasts between estimates of baseflow help discern multiple sources of water contributing to rivers. *Hydrology and Earth System Sciences*, 18(1):15–30.

Cartwright, I. and Morgenstern, U. (2018). Using tritium and other geochemical tracers to address the "old water paradox" in headwater catchments. *Journal of Hydrology*, 563:13–21.

Costelloe, J. F., Peterson, T. J., Halbert, K., Western, A. W., and McDonnell, J. J. (2015). Groundwater surface mapping informs sources of catchment baseflow. *Hydrology and Earth System Sciences*, 19(4):1599–1613.

Kirchner, J. W. (2003). A double paradox in catchment hydrology and geochemistry. *Hydrological Processes*, 17(4):871–874.

McDonnell, J. J. and Beven, K. (2014). Debates—the future of hydrological sciences: A (common) path forward? A call to action aimed at understanding velocities, celerities and

residence time distributions of the headwater hydrograph. *Water Resources Research*, 50(6):5342–5350.

Mei, Y. and Anagnostou, E. N. (2015). A hydrograph separation method based on information from rainfall and runoff records. *Journal of Hydrology*, 523:636 – 649.

Saraiva Okello, A. M. L., Uhlenbrook, S., Jewitt, G. P., Masih, I., Riddell, E. S., and Van der Zaag, P. (2018). Hydrograph separation using tracers and digital filters to quantify runoff components in a semi-arid mesoscale catchment. *Hydrological Processes*, 32(10):1334–1350.

Su, C.-H., Costelloe, J. F., Peterson, T. J., and Western, A. W. (2016). On the structural limitations of recursive digital filters for base flow estimation. *Water Resources Research*, 52(6):4745–4764.

---

## Short Comment (SC5) · 27 Dec 2019

Dear Drs. Antoine Pelletier and Vazken Andréassian,

Thanks for the manuscript. The method is very interesting and the presentation is clear. After reading the discussion paper I have one question. I wonder did you check how many time steps are updated within each hydrologic year? And where are those time steps concentrated on the hydrograph (i.e., after the peaks, after effective rainfall inputted, during recession). I think with those information will help the users/readers to understand the algorithm behavior. Also, it might be an indicator of the validity of the model if fewer updates were presented during each hydrologic year since the update

breaks the water balance relationship.

Best regards, Yiwen Mei

---

## Author Comment (AC8) · 6 Jan 2020

We would like to thank Dr Y. Mei for his question. It is an opportunity to illustrate the algorithm behaviour.

Two update mechanisms are present in the algorithm: an upward one and a downward one. The upward update occurs once a year, where measured streamflow reaches it yearly minimum; but the downward update occurs on a variable number of time steps, when computed baseflow is greater than measured streamflow.

For the three example hydrographs shown in the article, we counted the average number of time steps in which the reservoir level is updated each year through the downward update mechanism: 3.5/year for Vair river, 4.2/year for Virene river and 4.8/year for Petit Thérain river. Such a small sample does not allow to conclude to a general increasing trend with BFI, only that we have on average around 4 updates per year. We think that it is a reasonable number that supports the validity of the model, as highlighted by Dr. Y. Mei.

The figures below shows the location of time steps where the reservoir level is updated. Downward updates are mainly located during low-flows that follow a quick recession after the high-flow season. It is used by the model to adapt to a particularly quick decrease of measured streamflow, quicker than the response of the quadratic reservoir.
* * *
[Figure]

**Fig. 1.** Separated hydrograph of Vair river in Soulousse-sous-Saint-Élophe. Red dots are points in which the reservoir level is updated.

[Figure]

**Fig. 2.** Separated hydrograph of the Petit Thérain river in Saint-Omer-en-Chaussée. Red dots
are points in which the reservoir level is updated.

[Figure]

**Fig. 3.** Separated hydrograph of Virène river in Vire-Normandie. Red dots are points in which the reservoir level is updated.

---

## Author Response (AR1)

**Authors' response to reviews**

Antoine PELLETIER, on behalf of co-authors

24th January 2020

**1 Response to reviews**

We would like to thank Pr. Romanowicz and Pr. Cartwright for reading our manuscript and for their careful and useful reviews. We also thank Pr. Beven and Dr. Ding for their short comments which contributed to the rich interactive discussion. Here are our response to the points raised by the debate.

**Optimisation criterion**

As it is highlighted in Pr. Romanowicz's comment, the optimisation criterion that we use to calibrate the parameters of the hydrograph separation algorithm – Pearson correlation between baseflow and cumulated effective rainfall – is not univocal enough to be a satisfying correlation criterion. Indeed, it is subject to pseudo-periodical oscillations caused by the annual hydroclimatic cycle; it is visible on figure 3, page 14. We chose to use Pearson correlation since it is quick to compute and easy to interpret; however, a more complex optimization criterion could be developed indeed to get more univocal results.

**Recharge as linear fraction of streamflow**

This issue was underlined by both referees, Pr. Romanowicz and Pr. Cartwright: the fact that the water inflow of the quadratic reservoir is a linear fraction of daily measured streamflow is a major hypothesis of the algorithm. This is a very crude estimate of aquifer/subsurface recharge, which is a far more complex process including water flow through banks of the river, soil water balance, vegetation, seasonality, etc. Solving the groundwater transmissivity equation in a theoretical framework of a shallow aquifer connected to a river shows that recharge in anything but a linear fraction of streamflow; and real configurations of river-aquifer interactions are even more intricate.

However, as highlighted by the exchange with Pr. Beven, we do not claim to present a physically-based hydrograph separation method. For such a purpose, we would need an explicit recharge model, that would add more hypotheses and parameters: an elaborate production function in such an imperfect, but objective, algorithm would be a disproportionate weapon. Therefore, the inflow function composed of a fixed linear fraction of daily streamflow can be regarded as a basic estimate of the quantity of water that bears the catchment memory.

In the revised version of the article, we added a clearer explanation about the inflow function of the reservoir.

**Notations in equations**

We use a different notation for the continuous and the discrete versions of a variable: $X(t)$ for the continuous one and $X_t$ for the discrete one. Integration of the continuous differential equations is

made through a Eulerian explicit scheme and equation on page 7, line 15 is intended to explain how continuous variables are made from discrete measurements; in the revised version of the manuscript, we replaced it by a clearer plain-text explanation that will avoid confusion with other variables in the algorithm. As suggested by Pr. Cartwright, we added a table of variables to help the readability.

**Correlation between parameters**

Two parameters need to be calibrated in the algorithm presented in article. We tried to find a simple relationship between catchments' characteristics and parameters, in order to remove one degree of freedom in the optimization process; but we did not manage to find one. Since it is an unsuccessful point, it is not detailed in the manuscript; it is only mentioned at the end of page 17. In the revised version of the manuscript, we gave more details about it and the hydrological interpretation of such a result.

**Hydrograph separation and hydrological processes**

The separation of streamflow in two components – the slow and the quick one, or more precisely the delayed and the non-delayed one – is artificial: there is generally a wider range of hydrological processes at stake that have different response times to hydroclimatic events. Having said that, it is still possible to set up an arbitrary barrier between the *not too delayed* components of streamflow and the *delayed enough* components.

But attributing the flow component to an explicit physical process seems to be risky with a conceptual separation method like the one we present in the manuscript. Pr. Cartwright highlights that we define baseflow as the groundwater component of streamflow and quick flow as the surface runoff component. This distinction is improper and partial, we agree that it is an oversimplification of the system. Ideally, we should take into account various intermediate stores of water, for which we do not want to make hypotheses about their water residence time.

Anyhow, process interpretation of a conceptual separation method is hazardous practice – this was highlighted too by the interactive discussion with Pr. Beven. In the revised version of the article, after explaining the various types of water sources and their response behaviour, we defined baseflow as the sum of delayed streamflow components, whatever source they are from. The interest of this component is that it bears inter-annual memory of a catchment. We added this discussion to the introduction section.

**Abstract**

We believe in short and catchy abstracts that are likely to be read entirely. However, we agree that some crucial elements are missing in the proposed version. We propose a new abstract, which is a little longer.

**Introduction and review section**

In the revised version of the article, we merged the two sections. As underlined by Pr. Cartwright, some recent references are missing in the literature review; we rectified the review by adding new citations.

**Yearly reservoir update**

In the presented algorithm, there are two ways the level of the reservoir can be updated: downward, when the computed baseflow is greater than measured streamflow; upward, once a year, at the yearly minimum point of measured streamflow. The first one is understandable – baseflow cannot be greater than total streamflow – and the second one is just practical: as highlighted by Pr. Cartwright, in humid temperate French climate, the hypothesis that streamflow is only composed of baseflow at its yearly minimum is questionable, even though most recent French summers were dry enough to support this assumption. However, we need an upward update mechanism to compensate the downward one, since it allows the algorithm to be constrained to a water balance condition – that can be summed up by $\beta = BFI$. Removing the yearly upward update adds a degree of freedom for the model calibration and it causes significant difficulty to optimize the values of parameters. We added this discussion in the revised version of the paper.

**Role of $\tau$ parameter**

The idea of the optimization criterion used to calibrate the parameters is the following: we try to correlate two estimates of the quantity of water available to bear the catchment memory. First, computed baseflow; second, medium-term or long-term cumulative effective rainfall – which we estimated through the Turc-Mezentsev formula. Parameter $\tau$ is the length of the cumulating period for effective rainfall, it is an estimate of the response time of the catchment. Unfortunately, it gave less consistent results than the computed values of baseflow index, due to more calibration difficulty.

**Validation using hydrochemical data**

We agree with Pr. Cartwright that using geochemical data would be a less questionable way to calibrate our hydrograph separation process than the hydroclimatic criterion that we use. Unfortunately, we do not have available enough geochemical data in the French catchments of the sample to carry out a reliable study. We mentioned this idea as a perspective.

**Limitations of the method**

The technique presented in the manuscript is designed to be used on non-anthropised catchments and perennial streams. As highlighted by Pr. Cartwright, these limitations need to be more explicitly exposed in the conclusion. When the value of streamflow is affected by human activity – pumping, regulation by dams, canals, etc. – the algorithm can be applied only if the anthropised fraction of streamflow is small enough. This is a classical check in rainfall-runoff modelling. The case of groundwater abstraction resulting in disconnection is different: it is still caused by human activity but it indirectly affects streamflow. In such a case, if the aquifer contribution to the river is zero; but in the algorithm, reaching an empty filtering reservoir could result in some artefacts. This is a clear practical limitation of the method. In the revised version of the manuscript, we mentioned these limits of the study.

**2   List of changes in manuscript**

| Page | Lines | Description |
|---|---|---|
| 1 | 1-8 | Longer abstract |
| 1 | 17-18 | Definition of baseflow |
| 2 | 9 | Merging introduction and review section |
| 2 | 10-26 | New section defining hydrograph separation |
| 2 | 27-30 | Re-writing some sentences of the introduction to review |
| 3 | 1-3 | Re-writing some sentences of the introduction to review |
| 3 | 5 | Re-writing without process interpretation |
| 3 | 12-13 | Adding new reference |
| 3 | 24-25 | Discussion about tracer-based methods |
| 3 | 25 | Adding new references |
| 4 | 15 | Adding new references |
| 4 | 24 | Adding new references |
| 4 | 29 | Re-writing without process interpretation |
| 5 | 21-22 | Adding new references |
| 6 | 9-10 | Minor rephrasing |
| 6 | 19 | Re-writing without process interpretation |
| 6 | 24 | Re-writing without process interpretation |
| 7 | 6-9 | Discussion about the reservoir inflow function |
| 7 | 19 | Mentioning the table of variables |
| 8 | 13-15 | Clearer description of the integration scheme |
| 8 | 21 | Clearer description of the integration scheme |
| 9 | 4 | Minor rephrasing |
| 9 | 13 | Minor rephrasing |
| 10 | 12-14 | Discussion about the yearly minimum update |
| 11 | 16 | Minor rephrasing |
| 11 | 7 | Addong missing $\beta$ in the algorithm |
| 12 | 17-18 | Clearer definition of the parameter $\tau$ |
| 14 | 1 | Minor caption rephrasing |
| 18 | 8 | Correcting a typing error |
| 18 | 22-25 | Clearer explanation about the correlation between BFI and $\tau$ |
| 23 | 1 | Adding figure reference |
| 23 | 3-5 | Adding conclusive remarks about the optimization criterion |
| 23 | 11-12 | Perspective of evaluating the method using hydrochemical data |
| 23 | 13-14 | Discussion about catchments affected by human activity |
| 23 | 18 | Adding the package reference with a DOI |
| 28 | All | Table of variables |

Table 1: List of changes made by authors in the manuscript. Page and line numbers refer to the `latexdiff` version of the manuscript, available below.

**3   latexdiff version of the manuscript**

This version of the manuscript is available below. Added content is typeset in blue and removed one is in red footnotes.

[revised manuscript text omitted]